# Lightweight Midas Touch: A Paired-learning Framework for Flexible Design of Antimicrobial Peptides

## Abstract

Antimicrobial peptides (AMPs) are promising potential therapeutic agents against drug-resistant pathogens, owing to their broad-spectrum activity and minimal risk of resistance. However, prevailing AMP generative paradigms privilege only the dyad of model and data: small-scale AMP data and the inductive bias of bespoke models jointly circumscribe the generated candidates. This tenacious coupling between training data and model parameters imposes a constraint on the exploration of AMP diversity. To alleviate this, we introduce a paired-learning framework, lightweight Midas touch (LMT), which has four components: (1) a primer set, (2) a mapping table, (3) a target set, and (4) a converter. During training, a mapping table explicitly associates the primers with the targets, and a lightweight converter is trained to internalize this mapping correspondence. During inference, the candidate distribution is steered by re-specifying the primers, avoiding the need to re-design the converter. By liberating the primer set and mapping table from the data-model dyad, the framework dissolves their erstwhile tight coupling and affords flexible control over generation. Comprehensive evaluations demonstrate that the proposed framework yields AMP candidates of markedly enriched diversity while attaining accuracy. Consequently, by delivering diverse and novel AMP candidates, LMT constitutes a potent strength in the global campaign against antimicrobial resistance.

## 1 Introduction

Antimicrobial resistance (Holmes et al., 2016; Czaplewski et al., 2016; Magana et al., 2018; Laxminarayan et al., 2013; Conlon et al., 2016) exacerbates global mortality incurred by microbial infections. This formidable challenge remains a persistent and vexing frontier within the pharmaceutical field. Consisting of short amino acid sequences, antimicrobial peptides (AMPs) (Silva et al., 2020; Chaudhary & Mahfouz, 2024; Ferrazzano et al., 2022) possess the characteristics that enable the termination of various bacteria via mechanisms such as disrupting cell membranes (Goldberg et al., 2025), immunomodulation (Magana et al., 2020), specific target binding (Cullen et al., 2015), and interference with metabolic processes (Mookherjee et al., 2020). The extensive exploration of AMPs from diverse sources, encompassing both natural and synthetic origins, has demonstrated their capacity to combat microbial pathogens (Fjell et al., 2012; Jangra et al., 2025). Their broad-spectrum activity and lower risk of antimicrobial resistance position them as effective candidates for the development of novel alternative therapies, mitigating the threat posed to the lives of patients and the economy of our society (Torres Salazar et al., 2024; Murray et al., 2022). The heightened imperative to discover and design more effective AMPs has thus been underscored.

Numerous strategies have been developed to investigate novel AMPs. Bioassay-guided approaches, such as chromatography and fluorescence screening (Yang et al., 2024), demonstrate high precision yet are constrained by their time-intensive and costly characteristic. Due to these limitations, large-scale implementation remains impractical and ineffective. Traditional bioinformatics methods, such as the basic local alignment search tool (BLAST (Altschul et al., 1990)), offer significant advantages in sequence comparison, potentially identifying AMPs. Despite its effectiveness, the flat filter style of this method restricts the diversity of its conclusions, which results in bottlenecks in the scope of

exploration. In contrast, the employment of an intelligent strategy or a versatile tool (Porto et al., 2017) can certainly better facilitate and accelerate the process of exploring AMPs.

To date, numerous artificial intelligence (AI) based methodologies have been devised for novel AMP design, which can be summarized into two branches: identification and generation, as shown in Figure 1 (A). Identification emphasizes determining whether a given peptide sequence exhibits antimicrobial activity. For instance, Ma et al. built three prediction models, Attention, LSTM, and BERT, to mine AMPs from the human gut microbiome (Ma et al., 2022). Wang et al. proposed an explainable deep learning model, EvoGradient, to identify and optimize AMPs from the human oral microbiome (Wang et al., 2025a). As for generation, AI-guided models are employed to generate novel AMP candidates, creating sequences with potential therapeutic properties from scratch. These generated candidates often require iterative validation with multiple identification models or other filter strategies (e.g., physicochemical properties or peptide structures) to ensure they meet desired activity, specificity, and safety criteria before the wet validation. For instance, Szymczak et al. proposed a VAE-based model, HydrAMP, for AMP generation. After creating multiple candidates, they adopted three identification models for preselection and ranking (Szymczak et al., 2023). Wang et al. proposed a diffusion-based model, Pepdiffusion, to generate AMP candidates and used molecular dynamics (MD) simulations for further filtration (Wang et al., 2025c).

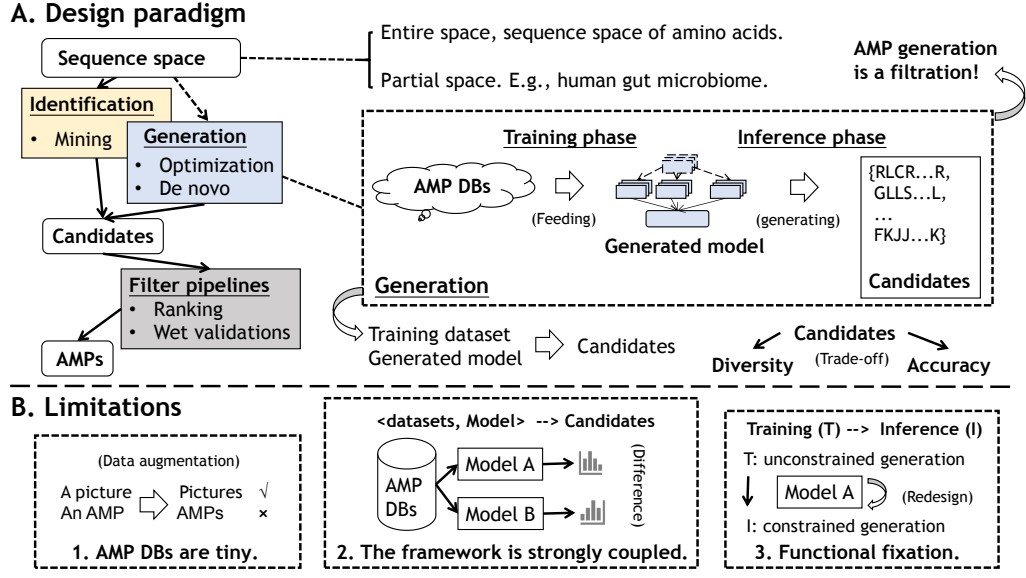

Figure 1: Process on designing AMPs and the limitations on the generation.

The dramatic reduction in both expenditure and turnaround time has rendered the investigation and refinement of AMP-generative models not merely significant but pivotal to the advancement of the antimicrobial field. However, there still remain several limitations, as shown in Figure 1 (B). First, AMP databases are tiny[1]. Meanwhile, direct data augmentation to swell the training dataset is non-rigorous, since a trivial alteration of the primary sequence can precipitate a profound and decisive shift in protein function, and only wet-lab experimental validation can determine whether a peptide is active or not. Consequently, generative models trained on modest corpora suffer from inherently limited generalizability. Second, the traditional generating framework is strongly coupled. The tight entanglement between data and model irrevocably predetermines the resultant candidates. Generative models tend to internalise the distribution of the training dataset, thereby producing novel instances that follow the same manifold. However, even if all models ostensibly learn the same distribution, each embodies a distinct inductive bias (Yun et al., 2020), yielding disparate outputs and divergent regions of feature space. As a result, the generative candidates thus bear the imprint of the training datasets and the model. Consequently, although a single generative model can produce

---

[1]The classical computer vision dataset *ImageNet* (Deng et al., 2009) contains over 14 million labeled samples, whereas the AMP database, *APD3* (Wang et al., 2016), has only 5099 peptides (last checked, Jan 2025).

an inexhaustible stream of peptides, genuine exploration of novel AMP candidates demands comprehensive and diverse generative samples, which are not limited to being generated from only one model. Third, functional fixation. The functionality of a model is fixed at the time of its construction. Consequently, the generated outputs are strictly bounded by these predetermined capabilities. Developing novel model architectures may serve as a provisional remedy, yet it is indisputably an arduous and technically exacting endeavor. Additionally, during the inference, sampling is performed in the latent space that encodes this distribution. The stochastic opacity of this latent sampling, however, deprives the controllability of designing AMP by targeted intent. For instance, while the generated candidates may follow the learned statistical traits of the training dataset, precise regulation of sequence length remains elusive, thereby diminishing the flexibility of AMP design pipelines that rely on generative architectures.

To this end, we propose Lightweight Midas Touch (LMT), a paired-learning framework conceived to empower the agile and controllable de novo design of AMPs. This framework is architected upon four constituents: (1) an explicit primer set, (2) a target dataset, (3) a mapping table, and (4) a lightweight converter. After establishing correspondences between the primer and target instances via the mapping table, LMT trains a converter to learn this mapping function. Both the primer set and the mapping table remain exogenous design levers, granting practitioners modular control over the generative objective. The primer set operates as a regulated conduit through which user-specified intent is infused, while the mapping table acts as a fulcrum whose minute perturbations propagate across the entire generative trajectory. For instance, even when the training dataset remains invariant, disparate mapping tables will yield distinct converters, thereby engendering divergent generative peptides. Meanwhile, with a fixed converter, the provision of different primer instances during inference will likewise elicit targeted, yet distinct, generated sequences, thereby enabling controllable, goal-oriented generation. LMT is both simple and efficient, affording governance over the generation process. Evaluations demonstrate that it confers exceptional plasticity upon the AMP generation, while simultaneously securing marked advantages in the diversity of the generated results.

## 2 LMT, A PAIRED-LEARNING FRAMEWORK

Generative paradigms rest upon two foundational pillars: a target dataset and a generative model. The model's overwhelming reliance on the target dataset ossifies its learned biases, constraining post-hoc variability and controllability. Such rigidity is detrimental in exploratory design tasks, where diversity is paramount. To alleviate it, besides the target dataset and the model (called converter in our framework), LMT introduces two additional constituents: a primer set and a mapping table, to serve as tunable control interfaces. These augmentations confer upon the entire generative architecture a flexible egress through which the output can be dynamically reconfigured.

### 2.1 CONSTITUENTS

LMT has four constituents, denoted as follows.

- A primer set, $\mathbb{L} = \{l_1, l_2, ..., l_n\}$, where the domain of its element is $\mathcal{L}$.

- A target dataset, $\mathbb{S} = \{s_1, s_2, ..., s_n\}$, where the domain of its element is $\mathcal{S}$.

- A mapping table, $m(.) : \mathcal{L}, \mathcal{S} \to \mathcal{M}$, where $\mathcal{M} = \langle \mathcal{L}, \mathcal{S} \rangle$.

- A converter, $f(l; \boldsymbol{\theta}|m(.)) : \mathcal{L} \to \mathcal{S}$, parametrized by $\boldsymbol{\theta}$.

To prevent a generative model from operating without guidance, LMT introduces a primer set as its governing substrate and, via a mapping table, couples it to the target set. Concurrently, a converter learns to internalize this mapping rule. Both the composition of the primer and the configuration of the mapping table afford freedom of choice. The two sets have the same number of elements, that is $|\mathbb{L}| = |\mathbb{S}| = n$. The mapping table is dedicated to connecting the elements between the primer and the target, and thus building a paired set $\mathbb{M} = \{\langle l_i, s_i \rangle\}_i$. The learnable converter is trained to generate predicted AMP candidates when fed with the customized primers, $f(\hat{l}; \boldsymbol{\theta}|\hat{l} \in \mathcal{L}) = \hat{s}$.

## 2.2 FRAMEWORK

As a transformative operator, the converter maps primer onto the target-domain manifold. Under this configuration, inter-module coupling is attenuated, thereby substantially enhancing the architecture's flexibility and extensibility. During the training phase, only a single fixed converter, coupled with a mapping table, yields a specialized model. Consequently, no modification of the model design strategy is required. Instead, a purposeful alteration of the mapping table alone suffices to instantiate models that generate qualitatively different outputs. During the inference phase, the primer constitutes the sole input to the model. Hence, generation can be flexibly steered through elementary, directional manipulations of the primers. The schematic diagram is shown in Figure2.

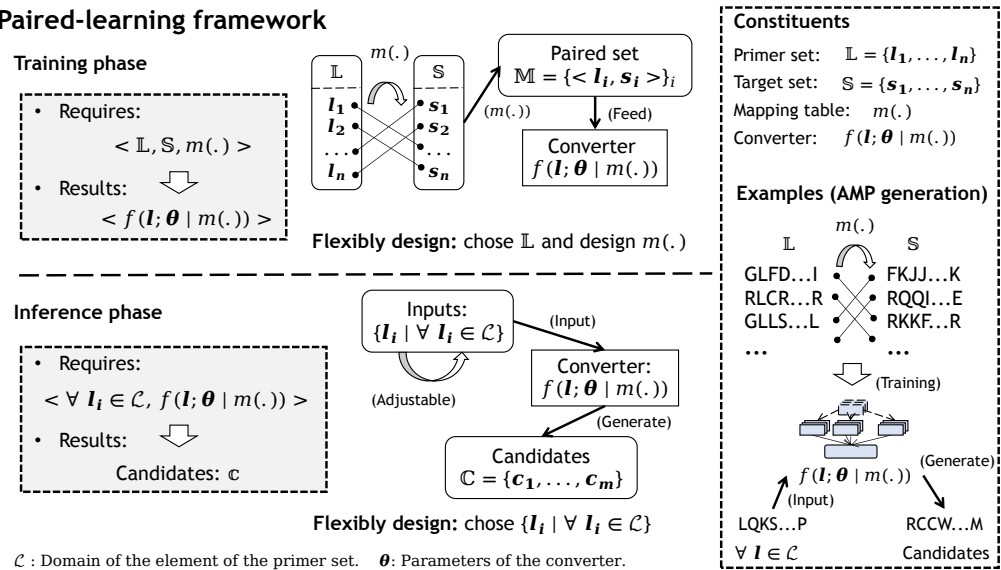

Figure 2: The framework of paired-learning.

The highlights of LMT are the primer set and the mapping table. Thus, this paper focuses on these two components, while fixing the converter as a *Transformer*[2] with two layers (Vaswani et al., 2017). We take verified AMPs as the target set and only consider 20 basic amino acids. To render the primer set explicit and manipulable, we here stipulate that its domain coincides exactly with that of the target set. Consequently, every element of the primer set is a sequence composed of the 20 standard amino acids. This prospect is exhilarating, for the paired-learning framework appears to transmute any arbitrary protein sequence into an AMP with effortless ease (Midas touch), dispensing entirely with the laborious pre-selection of mutational or evolutionary scaffolds. Meanwhile, it affords an unprecedented latitude for the flexible and unconstrained generation of AMP candidates.

**Polymorphic mapping table.** During training, the mapping table can be manipulated to instantiate models. The most elementary strategy is stochastic mapping, wherein each element of the target set is paired with a sequence drawn uniformly from the entire sequence space. This stochasticity amplifies inter-primer diversity: even when two targets are highly similar, their associated primers may exhibit pronounced divergence. This mapping disperses the manifold spanning the primer space onto the AMP field, thereby facilitating expansive exploration.

**Customized primers.** During inference, the generative candidates are steered by supplying specific primers. However, the efficacy of this control is contingent on the mapping table during training: the primer-target relations and latent constraints it encodes dictate what can be steered. For instance, any absence of length constraints in the mapping table will manifest as diminished controllability over sequence length at inference. Thus, controllability in inference is correlated with the choices encoded in the mapping table.

---

[2]Transformer models are proven as the universal approximators of sequence-to-sequence functions (Yun et al., 2020).

The preceding exemplifications are intended merely as an inaugural stimulus, and the design potential is limitless. The intrinsic virtue of the framework inheres in the latent plasticity of its constituents, which collectively furnish an expansive substrate for any conceivable design paradigm. Consequently, investigators are relieved of the burdensome imperative to devise novel architectures by modulating the training-stage mapping table and curating inference-stage primers, which enables convenient navigation of the sequence space with conceivable direction, allowing for the flexible orchestration of AMP generation. LMT constitutes a more general learning framework in which the selection of the primer set and the mapping table enjoys considerable degrees of freedom. Indeed, particular restrictions of these components can transform LMT into several well-known architectures, including both diffusion models (Ho et al., 2020; Kingma et al., 2021) and language models (Devlin et al., 2019; Vaswani et al., 2017) (see Appendix B for details).

**Converter.** LMT takes a two-layer canonical encoder-decoder Transformer (Vaswani et al., 2017) as its converter. Each layer comprises a Multi-Head Self-Attention sub-layer and a position-wise feed-forward network (FFN), arranged in a post-layer-norm configuration with residual connections and dropout probability . Each sub-layer is wrapped in a post-layer-norm residual connection accompanied by dropout, and sinusoidal positional encodings are added to the input embeddings. The decoder inserts an additional Masked Multi-Head Attention to prevent attending to future tokens. The final decoder states are linearly projected to logits of dimension 23 (vocabulary size) and passed through a softmax layer to yield per-position amino-acid probabilities.

The entire process begins with embedding. The primers and targets are tokenised at the residue level, mapped to 23-class integer indices (20 standard amino acids and 3 special characters), and padded to a common length. The encoder embeds these indices into $d_m$-dimensional vectors, scales them by $\sqrt{d_m}$, adds fixed sinusoidal positional encodings, and passes the result through two identical encoder layers. Each layer comprises multi-head self-attention with a padding mask that prevents attention to the padded positions, followed by a $d_f$-hidden-unit FFN with ReLU activation; both sub-layers employ post-layer-norm residual connections and dropout. The decoder receives a right-shifted copy of the target sequence, applies masked self-attention, encoder-decoder cross-attention, and the same FFN structure in two identical layers, and then outputs logits over the 23-token vocabulary. The results are compared with a label-smoothed distribution via KL-divergence summed over non-padding positions and normalised by the number of valid tokens (LabelSmoothingKLDivLoss). Throughout the process, the parameters are updated by Adam coupled to the Noam learning-rate schedule. Gradients flow back through the entire stack, updating all self-attention, feed-forward, layer-norm, and embedding parameters jointly. Detailed hyper-parameter configurations are provided in Section 3.1.

## 2.3 ANALYSIS

By regulating both the primer set and the mapping table, LMT exerts control over the converter and the generative candidates. Therefore, without loss of generality, we posit that $I(L; S) > 0$, where $I(\cdot)$ denotes *mutual information* (MI). This inequality entails that $L$ and $S$ are not independent, so that an observed $L$ carries non-zero information for the generation of $S$. Experimental assessments presented in Section 3 and Appendix E jointly corroborate the tenability of this postulate: the deployment of distinct test primers at inference, or the utilization of disparate mapping tables during training, invariably engenders non-negligible fluctuations in the distribution of generated sequences.

Under the traction of the primer set, the generated sequences become steerable; yet, the pivotal point is how to prescribe the construction of the test primers. Lemma 1 furnishes the requisite guidance.

**Lemma 1.** *Let $L$ be the random variable of the primer and $S$ be the random variable of the generative sequence. It holds that $H(S) \leq H(L)$, where $H(\cdot)$ is entropy.*

*Proof.* Recall that $H(S) - H(L) = H(S \mid L) - H(L \mid S)$ (see Appendix C for details). The conditional entropy $H(L \mid S)$ describes how much information entropy remains for the random variable $L$ given the value of the second random variable $S$. LMT aims to transform the primer space, which spans the entire sequence space, into the target space representative of AMPs. Concretely, for any given primer, the generative sequence is, with high probability, determined. Conversely, for a given generated sequence, its corresponding primer cannot be unambiguously identified, because multiple distinct primers may map to the identical sequence. Thus, it holds that $H(L \mid S) \geq H(S \mid L)$.

Finally, we can get

$$H(S) - H(L) = H(S \mid L) - H(L \mid S) \leq 0 \tag{1}$$

$\square$

This lemma asserts that the entropy of sequences generated from LMT is governed by that of the test primers. If the primers are confined to a narrow region of sequence space, the resultant candidates may collapse into a cluster of mutually similar sequences. Consequently, pairwise similarity among candidates rises while diversity plummets. An extreme illustration is a test primer set whose elements are identical or differ at only a handful of residues; the generated candidates may then be virtually indistinguishable from one another. Moreover, even when the primer sequences themselves are dissimilar, they may still be mapped to an identical sequence, thereby inducing redundancy among the generated candidates and further constraining the effective entropy of the outputs (see Appendix C for detailed assessments). This lemma furnishes actionable guidance. When the desideratum is a high-diversity AMP candidates, primers should be dispersed across sequence space. Conversely, if the investigative focus centres upon exploring variants of a single scaffold, primers may be confined to a narrow mutational radius. In either case, the test primers are advised to be co-optimized with the mapping table, rather than isolating them.

LMT exhaustively exploits the target dataset and, in so doing, gives rise to an indirect yet potent modality of data augmentation whereby it expands the primer set and distinct primers are tethered to identical target instances, thereby inflating the training data. For instance, a single paired training instance, $\langle l_1, s_1 \rangle$, can be augmented into two distinct instances by pairing the same target with a different primer $\langle l_1', s_1 \rangle$. Such augmentation, however, is not without peril: forcibly aligning heterogeneous loci of the full sequence space to the AMP manifold intensifies distributional overfitting. While this may marginally elevate generative accuracy, it simultaneously heightens the risk of overfitting and hence elevates sequence redundancy (see Appendix F for details).

## 2.4 DISCUSSION

LMT is well-suited to AMP generation. First, the AMP sequence is short, typically spanning merely 5 to 100 residues (Huang et al., 2023; Pinacho-Castellanos et al., 2021), and the database of validated peptides remains small in size. These attributes render feasible the use of lightweight converter models whose mapping tables can be iteratively refined within frequent, computationally frugal trainings. Second, AMP discovery is a filtration-driven exploration of sequence space, wherein both diversity and fidelity are of commensurate importance. By modulating the mapping table and, concomitantly, the primer set, the paired-learning framework furnishes an extensive repertoire of exploratory trials unshackled from the rigid interdependence of dataset and model that constrains conventional architectures, thereby amplifying the diversity of the generated candidates. In this way, the instantiation of a novel model demands no architectural redesign or algorithmic reconfiguration; a mere substitution of the mapping table suffices, guaranteeing the requisite flexibility for tailored design.

Then, we discuss how LMT mitigates the three limitations mentioned in the introduction. First, data scarcity. We propose an indirect data augmentation strategy. It does not require simulating similar pseudo-AMPs but only adding additional primer data. Moreover, noting that the distribution recovered by any learner is never an exact replica of the true, unknown distribution, it is, unavoidably, a composite of the latter and the inductive bias introduced by the model. As for LMT, the emergent distribution is co-determined by its four constituents. Consequently, it facilitates the continual instantiation of different models, broadens the AMP repertoire, and thereby accelerates the discovery of novel candidates. Simultaneously, it allows existing data to be exploited from multiple perspectives, enhancing data utility. Second, tight coupling. Traditional AMP generation frameworks are often confined to a dyad model-data framework. Data is fixed as AMP sequences, resulting in a strong dependency between the model and generated sequences. Monolithic models hinder the discovery of AMP diversity. LMT expands this dyad framework into a quaternary one by introducing primer sets and mapping tables, reducing the dependency of generated data on the model and thereby lowering coupling. Third, functional fixation. LMT alleviates the challenge of requiring model redesign to achieve new functions. Instead, new functionalities can be realized through simple adjustments to the training primer sets and mapping tables.

## 3 EVALUATIONS

We cover experimental setup in Section 3.1, present the results of LMT against its competitors in Section 3.2, and report the results of the variations within LMT in Section 3.3. Each model independently generated 10,000 non-redundant sequences for experimental evaluation. We use a single NVIDIA A40 48GB GPU for all experiments.

### 3.1 SETUP

**Baselines.** We consider three competitors, HydrAMP (Szymczak et al., 2023), Pepdiffusion (Wang et al., 2025c), and AMPDesigner (Wang et al., 2025b), represented by HDA, PDFS, and AMPDS, respectively. All of them follow their respective default configurations.

**Data.** AMP databases are constantly evolving and expanding. Consequently, it tends to exhaustively collect the latest dataset for training. However, it is difficult to ascertain whether superior predictive performance is primarily due to the innovative model or the enhanced quality of the training data, since a more comprehensive and diverse training data always engenders models of markedly superior generalization capacity. For the sake of fairness, we refrain from expanding the training corpus to its maximal breadth and instead restrict LMT exclusively to the APD3[3] (Wang et al., 2016). The three baselines are deployed exclusively in their originally released form, and no supplementary training or parameter updating is performed. In this way, LMT does not hold any advantage at the dataset level.

**Metrics.** We adopt two universal pairwise diversity scores, *Levenshtein Distance* (Levenshtein, 1966) and *PTIB Diversity* (Gao et al., 2023), denoted as $D_L(.)$ and $D_P(.)$, respectively. The overall diversity score is

$$D_k(\mathbb{S}_1, \mathbb{S}_2) = \sum_{i \in \mathbb{S}_1, j \in \mathbb{S}_2} \frac{D_k(i,j)}{m_1 \cdot m_2} \tag{2}$$

where $k \in \{L, P\}$, $|\mathbb{S}_1| = m_1$, and $|\mathbb{S}_2| = m_2$. We adopt three predictors, ATT, LSTM, and BERT, proposed by Ma et al. (Ma et al., 2022), to determine whether the generated sequence is an AMP[4]. For any generated sequence, the predictors yield a scalar in the interval $[0, 1]$. Adopting a threshold of $0.8$ in the following experiments (more results from different thresholds are shown in Appendix G), we classify a sequence as an AMP whenever its predicted value meets or exceeds this cut-off. Then, two evaluation criteria are employed: the intersection accuracy, whereby a sequence is deemed positive only if all three predictors concur in classifying it as an AMP, and the union accuracy, whereby a sequence is deemed positive if any single predictor classifies it. In this paper, the former is employed as the evaluative benchmark.

**Parameters.** The converter is instantiated as a two-layer Transformer with sinusoidal positional encoding, each with a model dimension of $d_m = 512$ and a feed-forward inner size of $d_f = 2048$. Multi-head attention is split into $h = 8$ heads. Dropout is set to $p = 0.1$ throughout the network. The optimizer is performed with Adam ($\beta_1 = 0.9, \beta_2 = 0.98, \epsilon = 1e - 9$) encapsulated within the NoamOpt schedule (the warm-up parameter is set to $4000$). The loss function fixed as *LabelSmoothingKLDivLoss* (Vaswani et al., 2017). Training epoch is set to 1000.

Table 1: Comparison of diversity and accuracy (baselines)

| Diversity | APD3 | HDA | PDFS | AMPDS | $LMT_{ran}$ | $LMT_{tmp}$ |
|---|---|---|---|---|---|---|
| Levenshtein Distance | 31.6027 | 18.3183 | 22.2459 | 15.9019 | **49.4902** | 28.3936 |
| PTIB Diversity | 37.2425 | 19.9994 | 25.2341 | 18.6140 | **59.6897** | 34.2525 |
| Intersection Accuracy | 0.7437 | 0.1656 | 0.3042 | 0.8166 | 0.6443 | **0.9040** |

---

[3]Data collection was completed in August 2024.

[4]Numerous studies have successfully mined AMPs from diverse datasets by means of these three predictors (Chen et al., 2024a; Xu et al., 2024), thereby attesting to the pipeline's credibility.

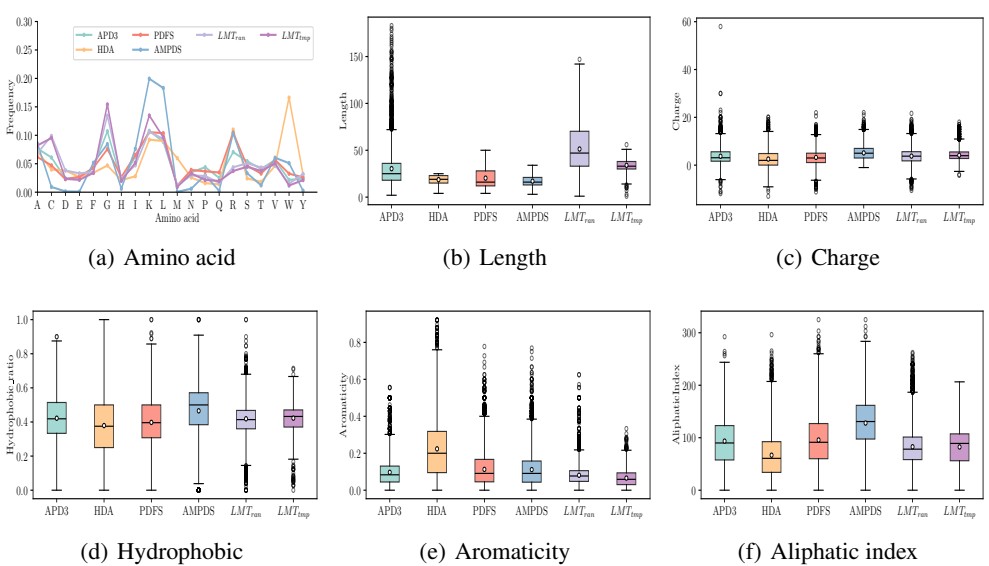

Figure 3: Comparison of physicochemical properties (baselines)

## 3.2 COMPARING WITH COMPETITORS

We build a random mapping table with a default random seed of 37, and train a converter. Upon this foundation, we constructed two distinct test primer sets for evaluation: a randomly assembled primer set (denoted as $\mathbb{L}_t$) and a tailor-made primer set[5]. Based on it, we get different generated sequences, denoted as $LMT_{ran}$ (serving as an anchor in all subsequent experiments) and $LMT_{tmp}$, respectively. The evaluations are presented as follows.

**Physicochemical properties.** We used an analysis tool from modlAMP (Müller et al., 2017) to visualize the physicochemical properties of all evaluated generative models, as shown in Figure 3. The results reveal marked inter-model divergence in the distributions generated by the distinct generative models. Notably, the distribution induced by $LMT_{ran}$ closely mirrors that of the training dataset, indicating that the model has internalized the latent AMP manifold and can generate sequences of comparable distributional fidelity via distinct primers.

**Diversity and accuracy.** Table 1 reports diversity scores. In comparison with its competing approaches, $LMT_{ran}$ exhibits markedly elevated sequence diversity. Even when the test primer set is subjected to stringent constraints, $LMT_{tmp}$ sustains a diversity level that surpasses all extant counterparts. Thus, LMT possesses a pronounced advantage in the generation of novel AMP candidates. Table 1 also presents the accuracy evaluations. The predictors yield an accuracy of 74.37% on the training set, indicating that the evaluation protocol still suffers from an inherent false-negative rate. We also randomly sampled 0.1M sequences uniformly from the entire sequence space of lengths from 10 to 100 and evaluated them using three independent predictors, denoted as Ran. The results indicate that the probability that a randomly assembled peptide with antimicrobial activity is found to be approximately 1.02% (see Appendix D for the details), establishing a lower bound: any generative model whose predicted accuracy exceeds this threshold is deemed effective. $LMT_{tmp}$ attains a high accuracy of 90.40%, markedly surpassing the bound and confirming its validity. Meanwhile, the results further reveal an inherent trade-off between diversity and accuracy in AMP generation. While AMPDS exhibits elevated accuracy, its diversity remains limited. Conversely, $LMT_{tmp}$, although less diverse than $LMT_{ran}$, surpasses it in accuracy. Crucially, $LMT_{tmp}$ exceeds all baselines with respect to both diversity and accuracy, thereby achieving an optimal balance between the two desiderata.

---

[5]The length of primers in the test set is confined between 30 and 40.

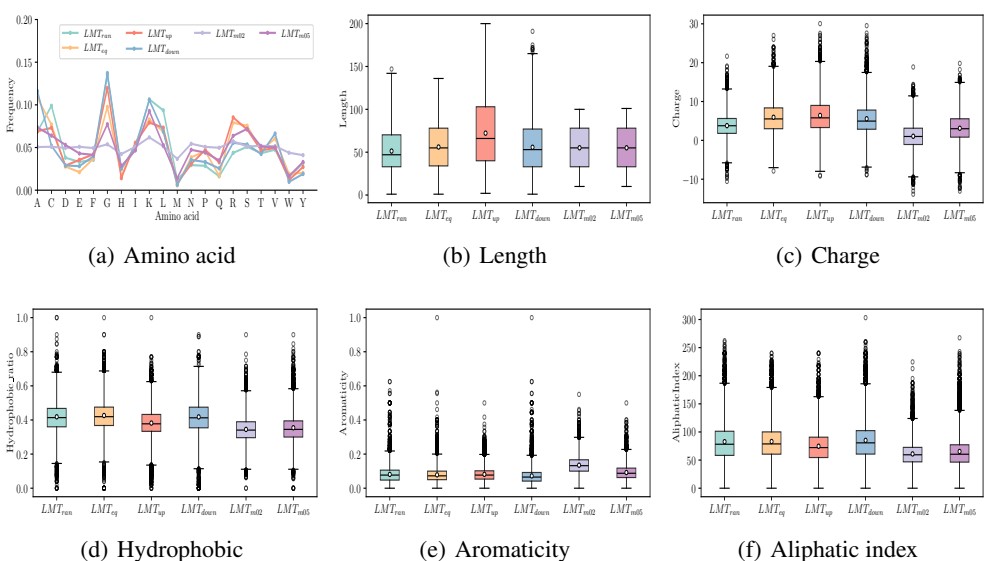

Figure 4: Comparison of physicochemical properties (modifying mapping table)

Table 2: Comparison of diversity and accuracy (modifying mapping table)

| Diversity | $LM_{ran}$ | $LMT_{eq}$ | $LMT_{up}$ | $LMT_{down}$ | $LMT_{m02}$ | $LMT_{m05}$ |
|---|---|---|---|---|---|---|
| Levenshtein Distance | 49.4902 | 53.8928 | **58.1053** | 52.9237 | 55.6528 | 54.7463 |
| PTIB Diversity | 59.6897 | 64.2046 | **68.3683** | 63.2746 | 64.9129 | 64.4856 |
| Intersection Accuracy | **0.6443** | 0.6285 | 0.5936 | 0.5962 | 0.0317 | 0.2838 |

## 3.3 VARIATIONS

In this part, we examine the individual consequences of (1) modifying the mapping table during the training phase and (2) varying test primers during inference.

**Results of modifying the mapping table.** During the training phase, we trained distinct converters by employing different mapping tables. During inference, the same test primer set ($\mathbb{L}_t$) was employed for sequence generation. All primers in the mapping table were stochastically sampled under the following constraints: for $LMT_{up}$ ($LMT_{down}$), every primer is shorter (larger) than its paired target, whereas for $LMT_{eq}$, a primer and its paired target share identical lengths. The primers of $LMT_{m02}$ and $LMT_{m05}$ were derived by masking the target set at rates of $0.2$ and $0.5$, respectively. Figure 4 demonstrates that the distributions of the generated peptides differ across models, corroborating that the choice of mapping table during training governs the statistical profile of the resultant antimicrobial candidates. The results underscore both the efficacy and the architectural flexibility of LMT. Table 2 quantifies the diversity and accuracy. All variants exhibit high diversity. Except for $LMT_{m02}$ and $LMT_{m05}$, all the rest attain competitive accuracy. $LMT_{m02}$ and $LMT_{m05}$, tasked with reconstructing full peptides from skeletal primers, interpret the randomized inference primers as putative backbones and merely mutate a handful of residues. Consequently, their outputs remain indistinguishable from random sequences. Moreover, lower masking ratios produce sequences increasingly similar to the inference primers, driving accuracy toward the random-peptide (predicted accuracy, $1.02\%$). Collectively, these results reveal an intricate relationship between the mapping table and the inference primers; the two must be co-designed within a unified framework rather than optimized in isolation.

**Results of varying the test primers.** During inference, varying the test primer set for $LMT_{ran}$ yields different generation distributions. All test primers are constructed by uniform random sampling under explicit length constraints: for $LMT_{tl10}$, the length of primers is restricted to $[10, 20]$; analogous rules are imposed on the remains, fixing the intra-set length span at exactly 10. As il-

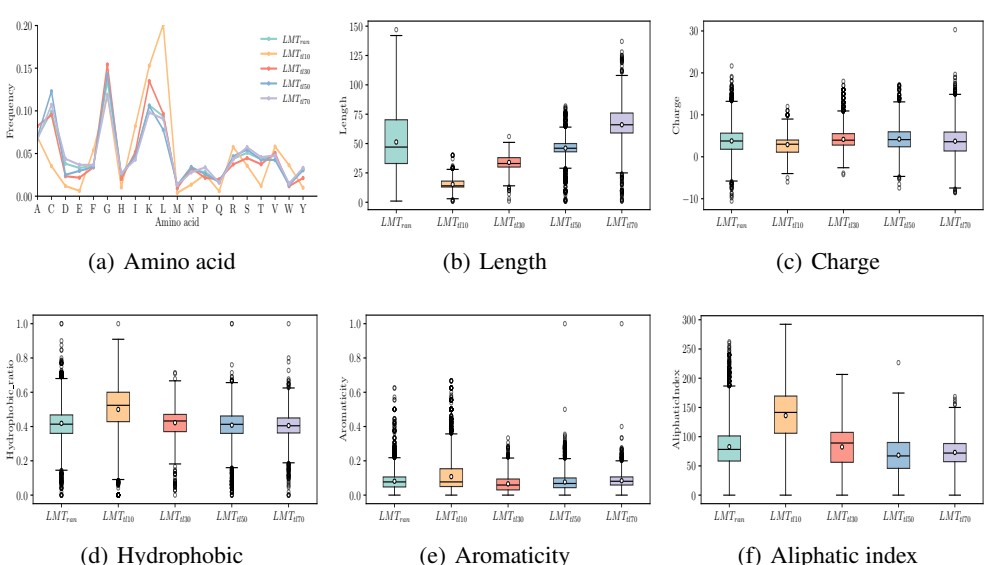

Figure 5: Comparison of physicochemical properties (varying test primers)

Table 3: Comparison of diversity and accuracy (varying test primers)

| Diversity | $LMT_{ran}$ | $LMT_{tl10}$ | $LMT_{tl30}$ | $LMT_{tl50}$ | $LMT_{tl70}$ |
|---|---|---|---|---|---|
| Levenshtein Distance | 49.4902 | 13.5655 | 28.3936 | 39.0923 | **56.9004** |
| PTIB Diversity | 59.6897 | 16.6258 | 34.2525 | 46.4426 | **67.7625** |
| Intersection Accuracy | 0.6443 | 0.8325 | **0.9040** | 0.6754 | 0.4552 |

lustrated in Figure 5, the deployment of distinct test primers yields generation distributions that are dissimilar. In particular, short primers bias the generator toward short-length candidates, whereas longer primers reorient the distribution toward sequences of greater span. Therefore, LMT affords control over the output repertoire through primer design, enabling on-demand tailoring of AMPs. Table 3 evaluates the diversity and accuracy. Shorter primers engender candidates that are narrower in diversity yet elevated in accuracy; conversely, longer primers expand sequence space at the cost of a modest decline in accuracy.

More experimental results, together with exhaustive particulars, are provided in the Appendix.

## 4 CONCLUSION

In this paper, we propose a paired-learning framework, LMT, that augments the conventional dyad of dataset and model with a primer repertoire and an explicit mapping table, thereby equipping the generative process with a tunable control interface, affording expansive dynamic flexibility. The framework is simple and efficient. Extensive evaluations demonstrate that it yields candidates that are markedly superior in diversity while preserving accuracy. By accelerating the discovery of novel AMPs with diversity, the LMT promises to expedite research on countermeasures against antimicrobial resistance.

## DECLARATION

We only use Kimi K2, a 1 trillion parameters mixture-of-experts language model with 32 billion activated parameters (Team et al., 2025), to aid writing.

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

## A   RELATED WORKS

AI-driven AMP discovery and design are currently advanced by two archetypes: discriminative models that predict antimicrobial activity or property from sequence, and generative models that generate novel sequences endowed with desired activities from scratch. Accordingly, it can be categorized into two aspects: mining from the valuable datasets or protein scaffolds, and generating from the latent space.

**Mining.** Equipped with pre-trained identification models, mining to check if a target sequence has antimicrobial activity is straightforward. Within this paradigm, the simplest approach is to screen the entire peptide sequence space. Huang et al. (Huang et al., 2023) mined the entire virtual library of peptides composed 6-9 amino acids to identify potent AMPs with an identifier. They synthesized the top-10 predicted peptides via solid-phase synthesis and tested their MIC values against *S. aureus*. All 10 peptides exhibited antimicrobial activities. This special pattern is attributed to the limited discrete entire space, which benefits from the relatively short sequence lengths. However, this method, although effective, is not optimistic in terms of resource consumption for exploring the entire space, especially as the sequence length increases. Therefore, mining from the partial sequence space is more desirable. Some candidate datasets with a greater likelihood of the presence of potential AMPs were favored (Chen et al., 2024a; Santos-Júnior et al., 2024; Ma et al., 2022; Xu et al., 2024). For instance, Ma et al. (Ma et al., 2022) mined antimicrobial peptides from the human gut microbiome, while Chen et al. (Chen et al., 2024a) mined a global marine microbial database. Both of them discovered novel AMPs successfully. The quality of this mining-driven mode hinges jointly on the candidate datasets and the identifiers employed. Selecting an inappropriate candidate dataset may prove futile. Furthermore, given that different identification models can produce varying results and no single model is universally superior, it is crucial to devise a suitable identification strategy prior to large-scale mining. Additionally, with the continuous emergence of new models, the results of mining-driven generation remain time-sensitive.

**Generation.** Different from the generative modes that rely on mining from the observable databases, some DNN-based models (Wan et al., 2022), such as variational autoencoders (VAEs) (Dean et al., 2021; Das et al., 2021; 2018; Wang et al., 2022), generative adversarial networks (GANs) (Lin et al., 2023; Van Oort et al., 2021; Cao et al., 2023; Yu et al., 2017), and diffusion models (Anand & Achim, 2022; Guo et al., 2024; Wu et al., 2024; Yi et al., 2023; Watson et al., 2023; Cao et al., 2024), generate novel candidates by sampling from the latent space. The latent space serves as a compact, continuous representation that encapsulates the intrinsic structure and variability of the training data, enabling the model to learn a more efficient and meaningful encoding of the data's underlying features. Take VAEs as an example, by first mapping the input data to a lower-dimensional latent space and then decoding it back to the original data space, these models produce a wide array of outputs while maintaining coherence and relevance to the learned data distribution. Sampling from the pre-learned latent space, VAE-based models, such as PepVAE (Dean et al., 2021), LSSAMP (Wang et al., 2022), and PepCVAE (Das et al., 2018), facilitate the generation of new samples analogous to the training data but with novel characteristics. In the case of GAN-based models, the generator learns to produce high-quality candidate sequences from the latent space through continuous correction and guided feedback from the discriminator. Lin et al. (Lin et al., 2023) synthesized 8 AMPs generated from their proposed GAN-based models, two of which exhibited broad-spectrum antibacterial effects and were effective against antibiotic-resistant bacterial strains, such as *S. aureus*. Diffusion-based models rely on adding noise to build the latent space, such as Gaussian noise, and then refining random noise into data samples by reversing a diffusion process. Finally, by sampling the latent space, the model can generate new elements that are similar to the training data. Based on this principle, Chen et al. proposed AMP-Diffusion (Chen et al., 2024b), Qi et al. proposed CDiffusion-AMP (Qi et al., 2024), and Wang et al. proposed Diff-AMP (Wang et al., 2024) for AMPs generation.

## B   RELATED VARIANTS

LMT renders both the primer set and the mapping table explicit, thereby allowing their manipulation to modulate generation. In doing so, the strong coupling between training data and AI model is loosened, endowing the overall architecture with the flexibility demanded by exploratory design tasks. Indeed, some extant learning paradigms already contain the very constituents we designate as the primer set and the mapping table, albeit in latent form. However, their visibility is obscured

only by certain intrinsic constraints. Through elementary formal transformations, these frameworks can be seamlessly subsumed within the paired-learning paradigm; prominent examples include both the diffusion models (Ho et al., 2020; Kingma et al., 2021) and the language models (Devlin et al., 2019; Vaswani et al., 2017).

**Diffusion models.** By imposing appropriate constraints, LMT readily metamorphoses into the diffusion paradigm, which generates data by iteratively denoising a random noise input through a learned reverse process. In this case, the primers are noisy corruptions of their counterparts in the target set, and the converter function as the denoiser in diffusion models, stripping the injected noise to recover the original data. In detail, we parameterize the Gaussian encoder on the target set with mean $\mu(s^{(t)}) = \sqrt{\alpha_t}s^{(t-1)}$, and variance $\Sigma_t(s^{(t)}) = (1 - \alpha_t)I$. Let $s_i^{(0)}$ be an original element from the target set. Upon completion of the $t$-th step, we get its paired primer, $l_i$.

$$l_i = s^{(t)} = \sqrt{\bar{\alpha}_t}s_i^{(0)} + \sqrt{1 - \bar{\alpha}_t}\epsilon^{(0)} \sim \mathcal{N}(s^{(t)}; \sqrt{\bar{\alpha}_t}s_i^{(0)}, (1 - \bar{\alpha}_t)I) \tag{3}$$

where $\bar{\alpha}_t = \prod_{i=1}^{t} \alpha_i$ and $\epsilon^{(0)} \sim \mathcal{N}(\epsilon; 0, I)$. In this way, the mapping table is also fixed, that is, $\{\langle l_i, s_i^{(0)} \rangle\}_i$. And, the converter is trained as the denoiser to simulate the denoising process and generate data. Thus, the diffusion scheme emerges as a paired-learning whose primer set and mapping table have been circumscribed: the primer set is uniquely determined by the target set with a rule $r(.)$, denoted $l_i = r(s_i)$, and the mapping table fixes the one-to-one correspondence between them, expressed as $\{\langle r(s_i), s_i \rangle\}_i$. These constraints attenuate the functional autonomy of the primer set and the mapping table, relegating them to mere epiphenomena and thereby centering the entire framework on the converter and the target set alone. Consequently, the overall architecture remains coupled. The compression of the operable manifold stifles the paired-learning framework's capacity for exploring novel AMPs.

**Language models.** Intuitively, the LM paradigm aligns itself with the paired-learning framework with particular immediacy: whether instantiated as cloze testing or translation tasks, its structure maps cleanly onto the primer and target duality. However, akin to the diffusion formalism, it imposes constraints on both primer set and mapping table. In cloze testing, the primer set is derived by masking designated tokens within the target set, while in translation tasks, cross-lingual semantic alignment enforces a rigid one-to-one correspondence. In short, both instantiations can be subsumed under the unified formalism, $\{\langle r(s_i), s_i \rangle\}_i$. This signifies that the primer set and the mapping table are determined by the target set. In the AMP design scenario, for instance, protein language models that primarily optimize sequences, e.g., via localized evolution or mutation (cloze testing), infill the high-quality, yet incomplete, scaffold to yield candidate AMPs. Nevertheless, these scaffolds must be painstakingly pre-engineered, rendering large-scale provision impractical. Moreover, the infilled sequences exhibit pronounced mutual similarity, thereby constraining generative diversity.

Extant architectures reveal conspicuous constraints once recast within the paired-learning framework: the primer set and mapping table are internalized by the target set and thereby immobilized. However, AMP generation is fundamentally an exploration of sequence space followed by requirement-driven filtration, where the diversity of the retained sequences is as critical as their fidelity, a priority different from the accuracy-centric imperative of translation tasks. This divergence furnishes an aperture for the manipulation of the primer set and the mapping table. By attenuating the constraints that bound the primer set and the mapping table from the target set, we confer upon LMT a expanded degree of freedom. In this way, during training, varying the mapping table diffracts the optimization trajectory into distinct generative pathways. During inference, targeted manipulation of primers steers the generation of desired sequences. This loosened coupling enables agile exploration of the sequence space and fosters the production of diverse candidates.

## C  SUPPLEMENTARY PROOF OF LEMMA 1

Recall Lemma 1, $H(S) \leq H(L)$, where $L$ and $S$ are the random variables of the primer and the generative sequence, respectively. It furnishes actionable guidance that the entropy of sequences generated from LMT is governed by that of the test primers. To substantiate this lemma, we first derive $H(S) - H(L) = H(S \mid L) - H(L \mid S)$, as follows.

$$H(S) - H(L) = -\sum_s \log p(s)p(s) + \sum_l \log p(l)p(l) \tag{4}$$

$$= -\sum_s \log p(s)(\sum_l p(l,s)) + \sum_l \log p(l)(\sum_s p(l,s)) \tag{5}$$

$$= -\sum_{l,s} p(l,s) \log p(s) + \sum_{l,s} p(l,s) \log p(l) \tag{6}$$

$$= \sum_{l,s} p(l,s) \log \frac{p(l)p(l,s)}{p(s)p(l,s)} \tag{7}$$

$$= \sum_{l,s} p(l,s) \log \frac{p(l,s)}{p(s)} - \sum_{l,s} p(l,s) \log \frac{p(l,s)}{p(l)} \tag{8}$$

$$= \sum_{l,s} p(s)p(l \mid s) \log p(l \mid s) - \sum_{l,s} p(l)p(s \mid l) \log p(s \mid l) \tag{9}$$

$$= -\sum_s p(s)H(L \mid S = s) + \sum_l p(l)H(S \mid L = l) \tag{10}$$

$$= H(S \mid L) - H(L \mid S) \tag{11}$$

The conditional entropy describes how much information entropy remains for the first random variable given the value of the second random variable. LMT aims to transform the primer space, which spans the entire sequence space, into the target space representative of AMPs. Concretely, for any given primer, the generative sequence is, with high probability, determined. Conversely, for a given generated sequence, its corresponding primer cannot be unambiguously identified, because multiple distinct primers may map to the identical sequence. Thus, it holds that $H(L \mid S) \geq H(S \mid L)$. Finally, we can get $H(S) - H(L) \leq 0$. The lemma is proved.

To further corroborate the derived entropy bound, we conducted an evaluation by varying the composition of the test primers. The result is summarised in Table 4, with the primer configurations specified below. Each test set comprises 10000 primers. All stochastic seeds are fixed at 37, and inference was executed with $LMT_{ran}$.

- $LMT_r$, sampled uniformly from the entire sequence space of lengths from 10 to 100.
- $LMT_{sim}$, 2000 primers sampled from the full space. Then, each of them mutated at $10\%$, independently repeated 4 times. In total, 10000 primers.
- $LMT_{rep}$, 2000 primers sampled from the sequence space and copied them 4 times.

Table 4: Entropy comparison between the test primers and their generations

| Entropy | $LMT_r$ | $LMT_{sim}$ | $LMT_{rep}$ |
|---|---|---|---|
| Test primer | 13.2877 | 13.2862 | 10.9657 |
| Generations | 13.0211 | 12.9761 | 10.8820 |

$LMT_r$ disperses primers, affording broad coverage of primer space. In contrast, $LMT_{sim}$ and $LMT_{rep}$ are more concentrated and contain duplicate primers. Consequently, the entropy of the test primer of $LMT_r$ exceeds that of the other two. Evaluation shows that, in every scenario, the entropy of the generative candidates never surpasses that of their corresponding test primers, again corroborating Lemma 1.

# D ASSESSMENTS ON APD3 AND RANDOM SEQUENCES BY THREE AMP PREDICTORS

We use three AMP predictors Ma et al. (2022), ATT (Attention for short), LSTM, and BERT, to evaluate their estimation accuracy on the AMP's dataset, APD3, as shown in Table 5. This evaluation

reveals that these three predictors still exhibit false-negative rates. When the threshold is set to $0.8$, the predictive intersection accuracy attains $74.37\%$. We also test the estimated accuracy of these baselines on the random sequence dataset by randomly sampling 0.1M sequences uniformly from the entire sequence space of lengths from 10 to 100, denoted as Ran, and evaluated them by these three independent predictors, as shown in Table 6. Experimental results demonstrate that, when sequences are randomly sampled from the full length range of $10 - 100$ amino acids, only $1.02\%$ are predicted to be AMPs when the threshold is set to $0.8$.

Table 5: Predictive accuracy on APD3

| Threshold | LSTM | ATT | BERT | Intersection | Union |
|---|---|---|---|---|---|
| 0.9 | 0.8966 | 0.7732 | 0.8119 | 0.6957 | 0.9374 |
| 0.8 | 0.9096 | 0.8231 | 0.8378 | 0.7437 | 0.9507 |
| 0.7 | 0.9172 | 0.8550 | 0.8507 | 0.7699 | 0.9603 |
| 0.6 | 0.9246 | 0.8825 | 0.8595 | 0.7936 | 0.9686 |
| 0.5 | 0.9313 | 0.8989 | 0.8654 | 0.8086 | 0.9736 |

Table 6: Predictive accuracy on 100000 random sequences

| Threshold | LSTM | ATT | BERT | Intersection | Union |
|---|---|---|---|---|---|
| 0.9 | 0.3302 | 0.0259 | 0.0635 | 0.0057 | 0.3551 |
| 0.8 | 0.3716 | 0.0408 | 0.0830 | 0.0102 | 0.4040 |
| 0.7 | 0.4000 | 0.0542 | 0.0974 | 0.0143 | 0.4385 |
| 0.6 | 0.4238 | 0.0676 | 0.1105 | 0.0186 | 0.4680 |
| 0.5 | 0.4458 | 0.0822 | 0.1228 | 0.0236 | 0.4953 |

# E  SUPPLEMENTARY RESULTS ON MODIFYING THE MAPPING TABLE

## E.1  IS THE MANIPULATION OF THE MAPPING TABLE GENUINELY INSTRUMENTAL?

Recall that we introduce three length-style LMTs in Section 3.3, $LMT_{eq}$, $LMT_{up}$, $LMT_{down}$. For $LMT_{up}$, every primer is shorter than its paired target sequence during the training phase. The remaining two exhibit analogous settings. We here evaluate whether this length rule is still kept during inference. Let $\triangle_>$ denote the count of sequences in the test primer set whose lengths exceed those of their corresponding generated sequences. $\triangle_<$ and $\triangle_=$ are defined analogously. $\triangle_{ratio}$, in turn, is defined as the quotient of the cumulative sum of length differences between each generated sequence and its paired primer sequence divided by the total number of elements in the test primer set. The result is shown in Table 7.

Table 7: Comparison of test primer length versus generated sequence length

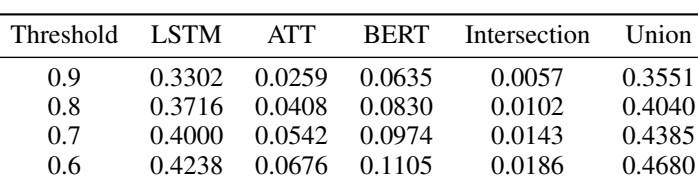

| LMTs | $\triangle_<$ | $\triangle_>$ | $\triangle_=$ | $\triangle_{ratio}$ |
|---|---|---|---|---|
| $LMT_{eq}$ | 6105 | 2257 | 1638 | 0.3591 |
| $LMT_{up}$ | 8388 | 1395 | 217 | 16.2014 |
| $LMT_{down}$ | 2344 | 6918 | 738 | -1.7030 |

Experimental results demonstrate that the length relationships encoded in the mapping table are reproduced during inference. For instance, in the $LMT_{up}$ mapping table, the lengths of sequences in the target set uniformly exceed those of their corresponding primers. This relationship is recapitulated at inference: among $10,000$ primers, $8,388$ generated sequences are longer than their paired primers. This indicates that the converter learns the length-mapping relationship and enables length-controllable generation via primer manipulation, underscoring the flexibility of LMT in AMP design. In the case of $LMT_{eq}$, although the number of length-congruent pairs is relatively lower,

its $\triangle_{ratio}$ value approaches zero, signifying that the deviation between primer length and generated sequence length remains at a small level. This observation also mirrors the length relationships encoded within the mapping table.

### E.2 MODIFYING RANDOM SEEDS

The mapping table of $LMT_{ran}$ is constructed by randomly pairing stochastically generated sequences with the target set, with the random seed fixed at 37. How about with distinct random seeds? We thus build multiple mapping tables with distinct random seeds and train the corresponding converters. For instance, $LMT_{r29}$ is trained by the mapping table with a random seed at 29. The same test primer set ($\mathbb{L}_t$) was employed for sequence generation. The results are as follows.

Figure 6 demonstrates that the distributions of the generated peptides still differ across models. It implies that even when the primer sets are sampled from an identical distribution, learning from disparate mapping tables can induce non-negligible distributional shifts in the generative sequences. Thus, LMT is distinguished by its inherent flexibility and variability. Table 8 quantifies the sequence diversity and accuracy of the generated candidates. All variants exhibit high diversity and competitive accuracy. Although the evaluated values exhibit fluctuation, such variability corroborates the capacity of LMT to be flexibly modulated for designing AMPs with pronounced diversity.

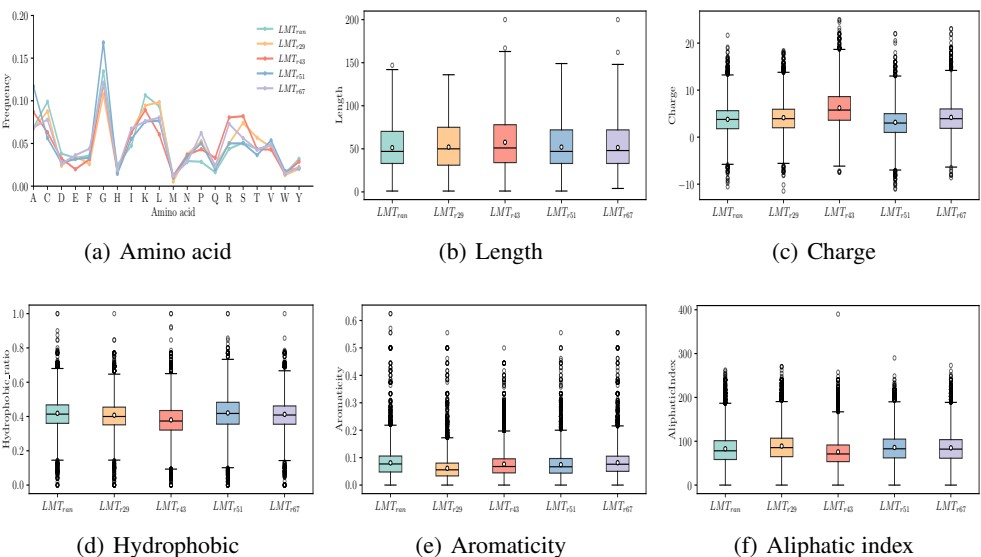

|         (a) Amino acid        |         (b) Length        |         (c) Charge        |
|:------------------------------|:--------------------------|:--------------------------|

|        (d) Hydrophobic        |        (e) Aromaticity        |        (f) Aliphatic index        |
|:------------------------------|:------------------------------|:----------------------------------|

Figure 6: Comparison of physicochemical properties (varying random seed)

Table 8: Comparison of diversity and accuracy (varying random seed)

| Diversity | $LMT_{r29}$ | $LMT_{r37}$ | $LMT_{r43}$ | $LMT_{r51}$ | $LMT_{r67}$ |
|---|---|---|---|---|---|
| Levenshtein Distance | 51.1531 | 49.4902 | 54.1699 | 49.8257 | 50.6038 |
| PTIB Diversity | 61.2445 | 59.6897 | 64.2708 | 59.7236 | 60.3716 |
| Intersection Accuracy | 0.6181 | 0.6443 | 0.5830 | 0.5877 | 0.6318 |

### E.3 REDUNDANCY RATES OF GENERATIVE SEQUENCES ACROSS DISTINCT MAPPING-TABLE CONFIGURATIONS

Distinct mapping tables engender different converters. At inference, we use a common primer set ($\mathbb{L}_t$) to assess the pairwise sequence redundancy among the generated sequences, with the results summarised in Table 9 and Table 10. If the distinct mapping tables exert no influence at all, fixing the target set would yield similar converters whose inferred distributions and sequence identity are indistinguishable, inevitably entailing pronounced redundancy among their respective outputs. The

assessments, however, reveal negligible pairwise redundancy (each below $3\%$), thereby corroborating the efficacy of LMT: the converters have indeed internalised distinct mapping relationships.

Table 9: Redundancy ratio across distinct LMTs (varying random seed)

| Duplicated Ratio | $LMT_{ran}$ | $LMT_{r29}$ | $LMT_{r43}$ | $LMT_{r51}$ | $LMT_{r67}$ |
|---|---|---|---|---|---|
| $LMT_{ran}$ | 1.0000 | 0.0225 | 0.0225 | 0.0232 | 0.0257 |
| $LMT_{r29}$ | 0.0225 | 1.0000 | 0.0206 | 0.0197 | 0.0225 |
| $LMT_{r43}$ | 0.0225 | 0.0206 | 1.0000 | 0.0215 | 0.0260 |
| $LMT_{r51}$ | 0.0232 | 0.0197 | 0.0215 | 1.0000 | 0.0244 |
| $LMT_{r67}$ | 0.0257 | 0.0225 | 0.0260 | 0.0244 | 1.0000 |

Table 10: Redundancy ratio across distinct LMTs (modifying mapping table, Section 3.3)

| Duplicated Ratio | $LMT_{ran}$ | $LMT_{eq}$ | $LMT_{up}$ | $LMT_{down}$ | $LMT_{m02}$ | $LMT_{m05}$ |
|---|---|---|---|---|---|---|
| $LMT_{ran}$ | 1.0000 | 0.0189 | 0.0204 | 0.0251 | 0.0000 | 0.0014 |
| $LMT_{eq}$ | 0.0189 | 1.0000 | 0.0206 | 0.0249 | 0.0000 | 0.0015 |
| $LMT_{up}$ | 0.0204 | 0.0206 | 1.0000 | 0.0199 | 0.0000 | 0.0010 |
| $LMT_{down}$ | 0.0251 | 0.0249 | 0.0199 | 1.0000 | 0.0000 | 0.0015 |
| $LMT_{m02}$ | 0.0000 | 0.0000 | 0.0000 | 0.0000 | 1.0000 | 0.0000 |
| $LMT_{m05}$ | 0.0014 | 0.0015 | 0.0010 | 0.0015 | 0.0000 | 1.0000 |

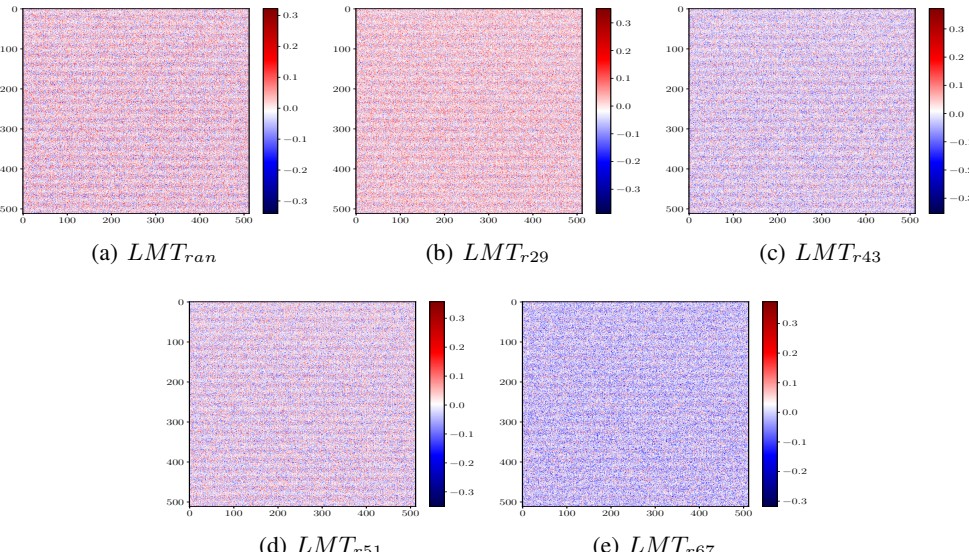

(a) $LMT_{ran}$      (b) $LMT_{r29}$      (c) $LMT_{r43}$

(d) $LMT_{r51}$      (e) $LMT_{r67}$

Figure 7: Weights of the multi-head attention in the last layer

### E.4 EVALUATIONS OF THE ENCODER WEIGHTS OF THE CONVERTER

Although preceding sections have repeatedly demonstrated that modulating the mapping table of LMT reliably steers the generated candidates, the evidence was necessarily acquired with a primer set. Meanwhile, we are particularly concerned with the stochastic mapping regime: primers are sampled uniformly from the entire sequence space and then randomly paired with elements of the target set, thereby maximising global entropy. Under this condition, the primer seems to be rarely constrained by the target set. Thus, will the encoder inside the transducer degenerate into an uninformative constant mapper? Therefore, to purge any residual influence of the primer set and to furnish conclusive evidence that the encoder of the transformer-based converter does not collapse into a trivial identity function, we isolate the projection layer of the final-block multi-head module in the last layer, *MultiHead(Q,K,V)*, extract its weights ($512 \times 512$), and render heatmaps for encoders

trained with divergent mapping tables. The results, as shown in Figure 7, reveal separated latent differences, thereby corroborating that the encoder internalises the mapping topology rather than jointly collapsing into a uniform, degenerate representation, especially for $LMT_{r29}$ and $LMT_{r67}$.

# F SUPPLEMENTARY RESULTS ON INDIRECT DATA AUGMENTATION

LMT has its own indirect data augmentation whereby it expands the primer set and distinct primers are tethered to identical target instances, thereby inflating the training data. For instance, a single paired training instance, $\langle l_1, s_1 \rangle$, can be augmented into two distinct instances by pairing the same target with a different primer $\langle l'_1, s_1 \rangle$. We take two kinds of mapping tables, random-based and mask-based, to test the data augmentation. $LMT_{ran}$ and $LMT_{m05}$ are tested as the benchmark (see Section 3 for detail). Their primer sets contain exactly the same number of elements as the target set. As for $LMT_{c2ran}$, the cardinality of its primer set is twice that of the target set, whereas for $LMT_{c3ran}$ the ratio is three-to-one. Their primer sets are randomly paired with the target sets, wherein each element of the target set appears twice in one case and three times in the other. The same principle governing on $LMT_{c2m05}$ and $LMT_{c3m05}$. In this way, not a single original AMP sequence is altered, yet the training set is enriched. The same test primer set ($\mathbb{L}_t$) was employed for sequence generation, and the evaluations are as follows.

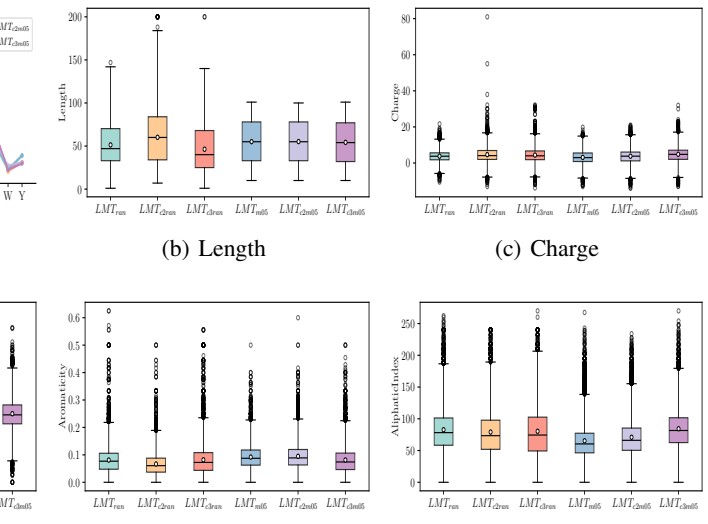

(a) Amino acid        (b) Length        (c) Charge

(d) Hydrophobic        (e) Aromaticity        (f) Aliphatic index

Figure 8: Comparison of physicochemical properties (data augmentation)

Table 11: Comparison of diversity and accuracy (data augmentation)

| Diversity | $LMT_{ran}$ | $LMT_{c2ran}$ | $LMT_{c3ran}$ | $LMT_{m05}$ | $LMT_{c2m05}$ | $LMT_{c3m05}$ |
|---|---|---|---|---|---|---|
| Levenshtein Distance | 49.4902 | 55.9929 | 47.9392 | 54.7463 | 54.5059 | 53.8147 |
| PTIB Diversity | 59.6897 | 66.0475 | 56.5646 | 64.4856 | 64.3785 | 63.6628 |
| Intersection Accuracy | 0.6443 | 0.6465 | 0.6981 | 0.2838 | 0.4214 | 0.4781 |
| Unique Ratio | 0.8791 | 0.6465 | 0.2159 | 0.9995 | 0.9915 | 0.8038 |

Figure 8 exhibits the physicochemical traits of the sequences generated under divergent regimes. Results reveal that models trained on mapping tables assembled with distinct construction styles generate discrepant sequence distributions. Table 11 quantifies the sequence diversity, accuracy, and unique ratio of the generated candidates. Unique ratio denotes the ratio of the number of non-redundant generated sequences to the total number of generative sequences. All variants exhibit high diversity. As the training data is augmented, the generated sequences exhibit monotonically increasing accuracy yet concomitantly diminishing unique ratio. Moreover, the amplitude of fluctuation under this trend is contingent upon the stylistic provenance of the mapping table: mask-based

mapping tables yield pronounced gains in accuracy while registering only modest increments in re-dundancy. Conversely, sequences generated under the random-based regime exhibit a opposed trend: their accuracy improves only marginally, whereas redundancy accumulates rapidly, culminating in an erosion of sequence uniqueness.

In summary, the data augmentation of LMT is not without peril: forcibly aligning heterogeneous loci of the full sequence space to the AMP manifold intensifies distributional overfitting. While this may marginally elevate generative accuracy, it simultaneously heightens the risk of overfitting and hence elevates sequence redundancy.

## G   PREDICTIVE ACCURACIES WITH DISTINCT THRESHOLDS

In this part, we evaluate the predictive accuracies of all generated sequences by the three predictors, ATT, LSTM, and BERT, with distinct thresholds, which serves as a supplement to the previous statement. Each of them contains 10,000 non-redundant generated sequences for the evaluation. The results are shown in Table 12 to Table 32.

Table 12: Predictive accuracy on HydrAMP sequences

| Threshold | ATT | LSTM | BERT | Intersection | Union |
|---|---|---|---|---|---|
| 0.9 | 0.2130 | 0.5103 | 0.4009 | 0.1254 | 0.6500 |
| 0.8 | 0.2807 | 0.5539 | 0.4425 | 0.1656 | 0.7022 |
| 0.7 | 0.3309 | 0.5840 | 0.4681 | 0.1967 | 0.7359 |
| 0.6 | 0.3692 | 0.6082 | 0.4888 | 0.2203 | 0.7624 |
| 0.5 | 0.4061 | 0.6286 | 0.5071 | 0.2451 | 0.7838 |

Table 13: Predictive accuracy on Pepdiffusion sequences

| Threshold | ATT | LSTM | BERT | Intersection | Union |
|---|---|---|---|---|---|
| 0.9 | 0.3652 | 0.5976 | 0.4776 | 0.2492 | 0.7237 |
| 0.8 | 0.4591 | 0.6480 | 0.5164 | 0.3042 | 0.7777 |
| 0.7 | 0.5206 | 0.6798 | 0.5421 | 0.3475 | 0.8084 |
| 0.6 | 0.5709 | 0.7052 | 0.5617 | 0.3810 | 0.8343 |
| 0.5 | 0.6146 | 0.7290 | 0.5803 | 0.4128 | 0.8550 |

Table 14: Predictive accuracy on AMPDesigner sequences

| Threshold | ATT | LSTM | BERT | Intersection | Union |
|---|---|---|---|---|---|
| 0.9 | 0.8174 | 0.8994 | 0.9288 | 0.7480 | 0.9811 |
| 0.8 | 0.8919 | 0.9226 | 0.9426 | 0.8166 | 0.9911 |
| 0.7 | 0.9217 | 0.9367 | 0.9479 | 0.8486 | 0.9942 |
| 0.6 | 0.9406 | 0.9464 | 0.9527 | 0.8717 | 0.9965 |
| 0.5 | 0.9537 | 0.9560 | 0.9580 | 0.8923 | 0.9974 |

Table 15: Predictive accuracy on $LMT_{ran}$ sequences

| Threshold | ATT | LSTM | BERT | Intersection | Union |
|---|---|---|---|---|---|
| 0.9 | 0.9280 | 0.6524 | 0.7700 | 0.6062 | 0.9540 |
| 0.8 | 0.9393 | 0.6911 | 0.7913 | 0.6443 | 0.9631 |
| 0.7 | 0.9462 | 0.7168 | 0.8042 | 0.6695 | 0.9692 |
| 0.6 | 0.9509 | 0.7377 | 0.8148 | 0.6902 | 0.9736 |
| 0.5 | 0.9550 | 0.7560 | 0.8222 | 0.7089 | 0.9758 |

Table 16: Predictive accuracy on $LMT_{eq}$ sequences

| Threshold | ATT | LSTM | BERT | Intersection | Union |
|---|---|---|---|---|---|
| 0.9 | 0.8584 | 0.6363 | 0.7895 | 0.5838 | 0.9156 |
| 0.8 | 0.8791 | 0.6811 | 0.8169 | 0.6285 | 0.9327 |
| 0.7 | 0.8898 | 0.7146 | 0.8290 | 0.6595 | 0.9409 |
| 0.6 | 0.8994 | 0.7411 | 0.8374 | 0.6844 | 0.9472 |
| 0.5 | 0.9085 | 0.7602 | 0.8457 | 0.7028 | 0.9529 |

Table 17: Predictive accuracy on $LMT_{up}$ sequences

| Threshold | ATT | LSTM | BERT | Intersection | Union |
|---|---|---|---|---|---|
| 0.9 | 0.8286 | 0.5906 | 0.7786 | 0.5431 | 0.9005 |
| 0.8 | 0.8502 | 0.6435 | 0.8016 | 0.5936 | 0.9188 |
| 0.7 | 0.8614 | 0.6738 | 0.8166 | 0.6214 | 0.9286 |
| 0.6 | 0.8712 | 0.6966 | 0.8265 | 0.6424 | 0.9362 |
| 0.5 | 0.8793 | 0.7209 | 0.8352 | 0.6638 | 0.9425 |

Table 18: Predictive accuracy on $LMT_{down}$ sequences

| Threshold | ATT | LSTM | BERT | Intersection | Union |
|---|---|---|---|---|---|
| 0.9 | 0.8135 | 0.6194 | 0.7398 | 0.5523 | 0.8893 |
| 0.8 | 0.8351 | 0.6671 | 0.7666 | 0.5962 | 0.9094 |
| 0.7 | 0.8502 | 0.6996 | 0.7830 | 0.6236 | 0.9243 |
| 0.6 | 0.8614 | 0.7239 | 0.7953 | 0.6464 | 0.9328 |
| 0.5 | 0.8698 | 0.7459 | 0.8050 | 0.6642 | 0.9404 |

Table 19: Predictive accuracy on $LMT_{m02}$ sequences

| Threshold | ATT | LSTM | BERT | Intersection | Union |
|---|---|---|---|---|---|
| 0.9 | 0.4124 | 0.0516 | 0.1304 | 0.0215 | 0.4532 |
| 0.8 | 0.4569 | 0.0758 | 0.1620 | 0.0317 | 0.5067 |
| 0.7 | 0.4849 | 0.0954 | 0.1841 | 0.0416 | 0.5403 |
| 0.6 | 0.5082 | 0.1151 | 0.2010 | 0.0502 | 0.5679 |
| 0.5 | 0.5302 | 0.1363 | 0.2166 | 0.0601 | 0.5958 |

Table 20: Predictive accuracy on $LMT_{m05}$ sequences

| Threshold | ATT | LSTM | BERT | Intersection | Union |
|---|---|---|---|---|---|
| 0.9 | 0.6631 | 0.3013 | 0.4985 | 0.2379 | 0.7445 |
| 0.8 | 0.6962 | 0.3596 | 0.5401 | 0.2838 | 0.7777 |
| 0.7 | 0.7188 | 0.3997 | 0.5642 | 0.3148 | 0.8017 |
| 0.6 | 0.7348 | 0.4311 | 0.5869 | 0.3410 | 0.8202 |
| 0.5 | 0.7496 | 0.4621 | 0.6041 | 0.3651 | 0.8347 |

Table 21: Predictive accuracy on $LMT_{tl10}$ sequences

| Threshold | ATT | LSTM | BERT | Intersection | Union |
|---|---|---|---|---|---|
| 0.9 | 0.9116 | 0.8454 | 0.9117 | 0.7876 | 0.9690 |
| 0.8 | 0.9286 | 0.8964 | 0.9226 | 0.8325 | 0.9789 |
| 0.7 | 0.9378 | 0.9219 | 0.9290 | 0.8558 | 0.9868 |
| 0.6 | 0.9470 | 0.9403 | 0.9334 | 0.8741 | 0.9903 |
| 0.5 | 0.9556 | 0.9516 | 0.9363 | 0.8878 | 0.9926 |

Table 22: Predictive accuracy on $LMT_{tl30}$ sequences

| Threshold | ATT | LSTM | BERT | Intersection | Union |
|---|---|---|---|---|---|
| 0.9 | 0.9790 | 0.9076 | 0.9391 | 0.8787 | 0.9935 |
| 0.8 | 0.9832 | 0.9284 | 0.9519 | 0.9040 | 0.9953 |
| 0.7 | 0.9849 | 0.9403 | 0.9587 | 0.9183 | 0.9966 |
| 0.6 | 0.9866 | 0.9485 | 0.9624 | 0.9290 | 0.9970 |
| 0.5 | 0.9880 | 0.9548 | 0.9653 | 0.9362 | 0.9976 |

Table 23: Predictive accuracy on $LMT_{tl50}$ sequences

| Threshold | ATT | LSTM | BERT | Intersection | Union |
|---|---|---|---|---|---|
| 0.9 | 0.9566 | 0.6888 | 0.7618 | 0.6427 | 0.9716 |
| 0.8 | 0.9630 | 0.7233 | 0.7833 | 0.6754 | 0.9774 |
| 0.7 | 0.9672 | 0.7454 | 0.7954 | 0.6955 | 0.9807 |
| 0.6 | 0.9699 | 0.7618 | 0.8049 | 0.7107 | 0.9829 |
| 0.5 | 0.9727 | 0.7801 | 0.8136 | 0.7274 | 0.9842 |

Table 24: Predictive accuracy on $LMT_{tl70}$ sequences

| Threshold | ATT | LSTM | BERT | Intersection | Union |
|---|---|---|---|---|---|
| 0.9 | 0.8963 | 0.4636 | 0.6446 | 0.4065 | 0.9264 |
| 0.8 | 0.9097 | 0.5140 | 0.6755 | 0.4552 | 0.9387 |
| 0.7 | 0.9189 | 0.5456 | 0.6938 | 0.4854 | 0.9470 |
| 0.6 | 0.9264 | 0.5725 | 0.7073 | 0.5102 | 0.9535 |
| 0.5 | 0.9318 | 0.5991 | 0.7195 | 0.5346 | 0.9582 |

Table 25: Predictive accuracy on $LMT_{r29}$ sequences

| Threshold | ATT | LSTM | BERT | Intersection | Union |
|---|---|---|---|---|---|
| 0.9 | 0.8690 | 0.6139 | 0.7976 | 0.5707 | 0.9216 |
| 0.8 | 0.8861 | 0.6620 | 0.8237 | 0.6181 | 0.9359 |
| 0.7 | 0.8944 | 0.6925 | 0.8382 | 0.6478 | 0.9433 |
| 0.6 | 0.9027 | 0.7162 | 0.8473 | 0.6704 | 0.9492 |
| 0.5 | 0.9097 | 0.7391 | 0.8559 | 0.6928 | 0.9548 |

Table 26: Predictive accuracy on $LMT_{r43}$ sequences

| Threshold | ATT | LSTM | BERT | Intersection | Union |
|---|---|---|---|---|---|
| 0.9 | 0.8276 | 0.5866 | 0.7651 | 0.5418 | 0.9035 |
| 0.8 | 0.8505 | 0.6325 | 0.7858 | 0.5830 | 0.9206 |
| 0.7 | 0.8641 | 0.6592 | 0.7975 | 0.6093 | 0.9313 |
| 0.6 | 0.8746 | 0.6801 | 0.8066 | 0.6288 | 0.9384 |
| 0.5 | 0.8834 | 0.7038 | 0.8141 | 0.6497 | 0.9454 |

Table 27: Predictive accuracy on $LMT_{r51}$ sequences

| Threshold | ATT | LSTM | BERT | Intersection | Union |
|---|---|---|---|---|---|
| 0.9 | 0.7976 | 0.6015 | 0.7382 | 0.5423 | 0.8658 |
| 0.8 | 0.8212 | 0.6504 | 0.7684 | 0.5877 | 0.8900 |
| 0.7 | 0.8357 | 0.6791 | 0.7861 | 0.6148 | 0.9030 |
| 0.6 | 0.8463 | 0.7032 | 0.7973 | 0.6352 | 0.9138 |
| 0.5 | 0.8550 | 0.7225 | 0.8069 | 0.6512 | 0.9225 |

Table 28: Predictive accuracy on $LMT_{r67}$ sequences

| Threshold | ATT | LSTM | BERT | Intersection | Union |
|---|---|---|---|---|---|
| 0.9 | 0.8410 | 0.6384 | 0.8004 | 0.5879 | 0.9079 |
| 0.8 | 0.8584 | 0.6836 | 0.8256 | 0.6318 | 0.9231 |
| 0.7 | 0.8696 | 0.7107 | 0.8413 | 0.6583 | 0.9342 |
| 0.6 | 0.8801 | 0.7348 | 0.8506 | 0.6803 | 0.9415 |
| 0.5 | 0.8881 | 0.7566 | 0.8574 | 0.7004 | 0.9459 |

Table 29: Predictive accuracy on $LMT_{c2ran}$ sequences

| Threshold | ATT | LSTM | BERT | Intersection | Union |
|---|---|---|---|---|---|
| 0.9 | 0.8414 | 0.6605 | 0.7861 | 0.5972 | 0.9090 |
| 0.8 | 0.8592 | 0.7095 | 0.8167 | 0.6465 | 0.9264 |
| 0.7 | 0.8696 | 0.7393 | 0.8311 | 0.6747 | 0.9345 |
| 0.6 | 0.8789 | 0.7628 | 0.8423 | 0.6977 | 0.9408 |
| 0.5 | 0.8886 | 0.7853 | 0.8507 | 0.7184 | 0.9483 |

Table 30: Predictive accuracy on $LMT_{c3ran}$ sequences

| Threshold | ATT | LSTM | BERT | Intersection | Union |
|---|---|---|---|---|---|
| 0.9 | 0.8823 | 0.7154 | 0.8142 | 0.6496 | 0.9362 |
| 0.8 | 0.8970 | 0.7620 | 0.8395 | 0.6981 | 0.9470 |
| 0.7 | 0.9057 | 0.7925 | 0.8546 | 0.7279 | 0.9559 |
| 0.6 | 0.9136 | 0.8158 | 0.8650 | 0.7513 | 0.9623 |
| 0.5 | 0.9189 | 0.8346 | 0.8729 | 0.7699 | 0.9678 |

Table 31: Predictive accuracy on $LMT_{c2m05}$ sequences

| Threshold | ATT | LSTM | BERT | Intersection | Union |
|---|---|---|---|---|---|
| 0.9 | 0.7743 | 0.4234 | 0.6385 | 0.3680 | 0.8434 |
| 0.8 | 0.8021 | 0.4811 | 0.6739 | 0.4214 | 0.8671 |
| 0.7 | 0.8167 | 0.5215 | 0.6934 | 0.4557 | 0.8825 |
| 0.6 | 0.8287 | 0.5530 | 0.7086 | 0.4844 | 0.8934 |
| 0.5 | 0.8399 | 0.5820 | 0.7227 | 0.5090 | 0.9029 |

Table 32: Predictive accuracy on $LMT_{c3m05}$ sequences

| Threshold | ATT | LSTM | BERT | Intersection | Union |
|---|---|---|---|---|---|
| 0.9 | 0.7862 | 0.4988 | 0.6749 | 0.4264 | 0.8619 |
| 0.8 | 0.8100 | 0.5561 | 0.7096 | 0.4781 | 0.8830 |
| 0.7 | 0.8248 | 0.5948 | 0.7309 | 0.5117 | 0.8971 |
| 0.6 | 0.8383 | 0.6253 | 0.7452 | 0.5379 | 0.9083 |
| 0.5 | 0.8512 | 0.6525 | 0.7567 | 0.5611 | 0.9194 |

