# OpenReview forum: "Lightweight Midas Touch: A Paired-learning Framework for Flexible Design of Antimicrobial Peptides"
_ICLR.cc/2026/Conference — Submitted to ICLR 2026_

### Official Review · Reviewer_WkGC · 2025-10-14

**Soundness:** 4
**Presentation:** 4
**Contribution:** 3
**Rating:** 6
**Confidence:** 3

**Summary:**

This paper introduces Lightweight Midas Touch (LMT), a paired-learning framework for antimicrobial peptide (AMP) design. Unlike traditional AMP generative models that tightly couple datasets and model parameters, LMT introduces two new components—a primer set and a mapping table—in addition to the target dataset and generative model. This design aims to decouple data from model biases, enabling flexible and controllable generation of diverse peptide candidates. The authors validate LMT against several state-of-the-art baselines (HydrAMP, PepDiffusion, AMPDesigner) using physicochemical and diversity metrics, reporting superior diversity and competitive accuracy. They also explore different configurations of the mapping table and primer set to demonstrate controllability and flexibility.

**Strengths:**

The paper presents a conceptually novel framework that generalizes several existing paradigms (e.g., diffusion and language models) under a unified formalism. The inclusion of a mapping table as a tunable control mechanism is elegant and well-motivated. The experimental evaluation is broad, covering diversity, accuracy, and controllability. Compared with existing models, the method achieves a good trade-off between peptide diversity and predictive performance. The theoretical discussion (e.g., Lemma 1 on entropy bounds) provides an interesting analytical perspective on diversity control. Overall, the paper offers a well-written and ambitious attempt to rethink controllable generative peptide design.

**Weaknesses:**

While conceptually intriguing, the practical novelty and biological significance of LMT are somewhat limited. The “paired-learning” idea largely formalizes an existing notion of conditioning or data pairing, and its empirical benefits are modest given the small dataset and limited validation (no wet-lab or structural confirmation). The theoretical analysis, though elegant, is superficial. Lemma 1 provides little practical insight beyond intuitive entropy control. Benchmarking lacks rigorous baselines such as protein-specific diffusion or autoregressive sequence models trained under similar data constraints. Moreover, the evaluation relies solely on computational predictors, which are known to produce false positives. Reproducibility is also questionable: the mapping table construction, random seeds, and parameter details are underspecified.

**Questions:**

1. The paired-learning framework resembles conditional generation or control through prompts in diffusion or transformer-based models. Authors should better clarify what new learning capability LMT introduces.
2. The results rely entirely on in silico AMP predictors (LSTM, BERT, ATT) with known bias; there is no experimental validation or external benchmark dataset.
3. The mapping table’s generation and training process are described conceptually but lack mathematical or algorithmic specificity—how is it optimized or regularized?
4. Theoretical analysis (e.g., Lemma 1) reads as a reformulation of intuitive properties, without empirical verification of information-theoretic claims.
5. While LMT yields “diverse” peptides, the biological or structural meaning of diversity (e.g., sequence motifs, charge distribution) is not examined.
6. Models like ProtGPT2 or ESM-IF could provide stronger baselines for peptide generation; excluding them weakens the comparative claims.
7. Important implementation details (mapping-table sampling, hyperparameters, training stability) are missing, making replication difficult.

---

> ### Author Response · Authors · 2025-12-03
> **Response to WkGC (1/2)**
>
> Thank you for the time and effort spent on reviewing our work. We appreciate your comments and address your concerns and questions below.
>
> Q1. We appreciate your insightful comment. LMT is grounded in paired-learning and constitutes a more generalised framework whose objective is to learn a mapping relationship from a primer set to a target set. In fact, this Mapping learning principle is ubiquitous in contemporary generative paradigms: diffusion models internalise the mapping between noise and original data, while LLMs learn the correspondence mapping between source and target languages. The distinctive contribution of LMT is to extend explicit mapping-based learning to latent, user-defined relationships that are not immediately observable in the raw data. In other words, not all mapping relationships are straightforward. The experimental results confirm the feasibility of this strategy: when a random primer set is coupled with a mapping table that enforces, for example, a length-upshift constraint (LMT_up), the generated sequences exhibit the same directional bias at inference.
> The approach offers two concrete advantages.  First, it endows latent-space sampling with a navigation mechanism. The primer acts as an explicit steering vector, permitting targeted traversal of the latent space rather than undirected random sampling. Second, it enables fine-grained control: instead of prompting a language model once and obtaining a homogeneous batch, LMT assigns a dedicated primer to each prospective sequence, thereby tailoring the generative act to the specificity of the individual requirements.
>
> Q2. We argue that numerous studies have successfully mined AMPs from diverse datasets by means of these three predictors. For instance, Xu et al. [6] mined AMPs from sludge, and Chen et al. [7] mined AMPs from the global marine microbial database. Indeed, all of the authors restricted themselves to employing these three predictors exclusively as predictive engines, without invoking any additional AI models. Thus, these models are widely acknowledged to constitute the present generation of premier predictors. I'm afraid that any predictor, no matter how impeccably engineered it is, remains intrinsically biased. Wet-lab validation thus remains the sole unassailable criterion of biological veracity. However, ICLR has recently accepted a series of studies confined entirely to purely in silico verifications [1-5], exemplified by the antibody design work of Wu et al [3]. These standards inspired our own submission.
>
> [1] CBGBench: Fill in the Blank of Protein-Molecule Complex Binding Graph. ICLR, 2025.
> [2] Steering Protein Family Design through Profile Bayesian Flow. ICLR, 2025.
> [3] A Simple yet Effective $\Delta\Delta G$ Predictor is An Unsupervised Antibody Optimizer and Explainer. ICLR, 2025.
> [4] ReNovo: Retrieval-Based \emph{De Novo} Mass Spectrometry Peptide Sequencing. ICLR, 2025.
> [5] Fine-Tuning Discrete Diffusion Models via Reward Optimization with Applications to DNA and Protein Design. ICLR, 2025.
> [6] Waste to resource: Mining antimicrobial peptides in sludge from metagenomes using machine learning. Environment International, 2024.
> [7] Global marine microbial diversity and its potential in bioprospecting. Nature, 2024.
>
> Q3. Thank you for your comment. We have incorporated an expanded part in the revised manuscript (Sections 2.2 and 3.1). We used the Adam optimizer with $\beta_1 = 0.9, \beta_2 = 0.98, \epsilon = 10^{−9}$, and varied the learning rate over the training steps, $(512^{-0.5} \cdot min(step^{-0.5}, step \cdot 4000^{-1.5}))$. As for the regularization, we set drop=0.1 and label smoothing equal to 0.1.
>
> Q4. Thank you for your comment. The detailed derivation of Lemma 1 is provided in Appendix C. Additionally, we conducted experimental verification. As shown in Table 4 (in Appendix C), the experimental results also corroborate Lemma 1.
>
> Q5. The charge diversity and other biological metrics of the sequences are shown in the physicochemical property distribution plots in this paper (e.g., Figures 3-6). Regarding motif analysis, such explicit statistical tests are rarely encountered in the AMP field: the inaugural publications on AMPDS, HDA, and PDFS included no motif enrichment experiments whatsoever. This omission is attributable chiefly to the intrinsically short length of AMPs. For instance, Huang et al. [8] restricted their generated sequences of 6–9 residues, whereas the most widely cited motif-discovery tool, MEME Suite (https://meme-suite.org/meme/tools/meme), imposes a hard lower bound of 8 amino acids. To satisfy this requirement, investigators must first cull all shorter sequences, thereby compromising the integrity of the generated data set.
>
> [8] Identification of potent antimicrobial peptides via a machine-learning pipeline that mines the entire space of peptide sequences. nature biomedical engineering, 2023.

---

> ### Author Response · Authors · 2025-12-03
> **Response to WkGC (2/2)**
>
> Q6. Some protein design models, for example, ProtGPT2 (a deep, unsupervised transformer‑based language model pretrained on UniRef50 for protein sequence generation and design), are not specifically designed for AMPs. ProtGPT2's core component is to capture the statistical patterns of natural amino acid sequences. A critical challenge arises in defining the inputs for the ProtGPT2 when the objective is to produce AMPs at scale. Certainly, endowed with a sufficiently informative and promising protein scaffold, it may well be positioned to devise a novel class of AMPs. However, unless further fine-tuning is performed, the large-scale production of AMPs is not feasible. Consequently, models that require task-specific fine-tuning before they can yield AMPs at scale were excluded from our comparative evaluation.
> Even so, we still employed the original ProtGPT2 to generate 10000 de novo non-redundant sequences for validation. The results and generation settings are summarised below.
>
> | Threshold | Att  | LSTM | BERT | Intersection | Union  |
> |-----------|--------------|---------------|---------------|--------------|--------|
> | 0.9       | 0.2389       | 0.004         | 0.0046        | 0.0009       | 0.2405 |
> | 0.8       | 0.2666       | 0.0054        | 0.0072        | 0.0015       | 0.2689 |
> | 0.7       | 0.289        | 0.0062        | 0.0095        | 0.0021       | 0.2917 |
> | 0.6       | 0.3099       | 0.0073        | 0.0103        | 0.0026       | 0.3128 |
> | 0.5       | 0.3264       | 0.0089        | 0.0117        | 0.003        | 0.3299 |
>
> Levenshtein Distance: 49.5673311
>
> PTIB Diversity: 55.48085128
>
> **Generation settings**:
> inputs: "<|endoftext|>";
> random seeds: 37;
> max_length=25,
> do_sample=True,
> top_k=100,
> repetition_penalty=1.2,
> length_penalty=1.2,
>
> Q7. We release the source codes and provide the relevant parameter settings. The codes are available in Supplementary Materials, and the detailed parameter settings are in Section 3.1.
>
> Thank you again for your time and comments.

---

### Official Review · Reviewer_6HzQ · 2025-10-29

**Soundness:** 2
**Presentation:** 2
**Contribution:** 2
**Rating:** 2
**Confidence:** 3

**Summary:**

The paper proposes a “paired-learning framework” (LMT) for antimicrobial peptide (AMP) generation. LMT comprises four components: a primer set, a target set (AMP dataset), a mapping table, and a lightweight converter (a two-layer Transformer). During training, primers and target sequences are paired via the mapping table to learn a mapping from primers to targets. At inference, changing the primer set is used to controllably steer the output distribution without redesigning the model architecture.

**Strengths:**

1. By making the primer and the mapping table explicit and manipulable, the framework provides a simple, implementable control channel. A small model can achieve distribution steering to some extent (e.g., sequence length distribution follows constraints imposed on inference primers).
2. In certain settings, the reported diversity metrics are significantly higher than baselines, and in some cases this coincides with higher predictive accuracy (e.g., LMTtmp intersection accuracy 0.904 vs. AMPDesigner 0.817).

**Weaknesses:**

1. In some indirect “data augmentation” settings (pairing multiple primers with the same target), accuracy increases but uniqueness decreases (redundancy rises). This suggests potential overfitting to the target distribution at the expense of novelty. The paper does not offer a systematic approach to balance control and diversity (e.g., penalties, deduplication strategies, sampling temperature, property coverage objectives).
2. Methodologically, the framework is essentially a seq2seq model that learns to map arbitrary primers to targets via forced pairing. With random mapping tables and semantically unrelated primers, training resembles memorizing a random code-to-target mapping. The mechanism enabling generalization to unseen primers and principled controllable generation is unclear and appears largely empirical (e.g., length controllability). Relative to established conditional generation, prompt-driven language models, or conditional diffusion, the novelty is more in presentation than in the core method.
3. Fairness of baseline comparison is questionable. Were all three baselines retrained on APD3 with equivalent settings? Many generative methods are originally trained on larger or different datasets; direct comparison may be unfair. Although the paper emphasizes using the same dataset, retraining details, ablations, and released code are missing, which hinders reproducibility and verification.
4. Key implementation details for the converter are insufficient: is it encoder–decoder or decoder-only? Loss, optimizer, learning rate, regularization, tokenization, positional encoding, sampling strategy, etc. are not specified beyond “default,” compromising reproducibility.
5. Reported “accuracy” relies entirely on the intersection of three deep learning predictors, and the authors acknowledge that these predictors yield only 74.37% intersection accuracy on APD3, implying substantial false positives/negatives. Predictor-based evaluation risks circular validation bias with the training distribution and does not convincingly demonstrate discovery of superior, truly effective AMPs.
6. No open-source code is provided, making reproduction difficult.

**Questions:**

Improve fairness and transparency in baseline comparisons. Retrain baselines under the same data and training budget with fully documented, reproducible protocols, and release code and configurations.

---

> ### Author Response · Authors · 2025-12-03
> **Response to 6HzQ (1/2)**
>
> Thank you for the time and effort spent on reviewing our work. We appreciate your comments and address the concerns you raised as follows:
>
> W1. Thank you for your insightful comment. That's really a good question. To the best of our knowledge, the data augmentation strategy described in this paper is unprecedented, at least in the field of AMP generation. Meanwhile, we have already clearly and explicitly highlighted this side-effect in our manuscript, redundancy.
>
> It must first be recognised that the redundancy is unavoidable. This arises because the domain of primers coincides with the entire sequence space spanned by the 20 standard amino acids, whereas the target set is the vastly smaller AMP sub-space. Therefore, it compresses a large space into a smaller one, inevitably resulting in multiple primer sequences mapping to the same AMP sequence, and thus generating redundant sequences. Our data augmentation procedure amplifies this phenomenon, thereby tightening the latent distribution around the antibacterial manifold. Precision improves, but the model simultaneously memorises the duplicated target and loses out-of-distribution flexibility.
>
> A straightforward strategy is: pre-sample additional primers, generate an oversized candidate pool, and apply deduplication post hoc. Although effective, this solution is not elegant. We also explored algorithmic solutions, referencing existing methods, and found this problem highly challenging: it seems difficult to resolve redundancy without sacrificing primer data. The key point is that one primer in LMT corresponds to a single target, not multiple. Taking penalty strategies as an example, unlike traditional language models that generate numerous candidate sequences from a single prompt, it can discard some low-quality generated sequences. LMT, of course, can employ token penalty algorithms to potentially filter out high-frequency tokens. But, even with algorithmic filtering, it still needs to directly eliminate duplicate sequences generated from the primers at the expression level, mirroring the same result of the above straightforward strategy. More primers are still required at the input level.
>
> Fortunately, apart from the experiments in Appendix F testing data augmentation effects, none of the other experimental settings utilized data augmentation mode. Yet, experimental results show LMT can still generate sequences with competitive accuracy. This implies that if users prioritize generating accuracy, they can adopt the data augmentation mode, where the redundancy effect is pronounced. Without the data augmentation, LMT still achieves high performance. After all, there's no such thing as a free lunch.
>
> W2. Thank you for your comment. LMT is presented as a learning framework rather than a single, frozen model. By simply redesigning the mapping table, the same backbone can be repurposed for new objectives without architectural redesign. While the mechanism can be viewed as a form of seq2seq learning, the emphasis is shifted away from mappings that are subjectively obvious, such as source-to-target sentences or noise-to-clean images, toward relationships that are latent and initially unspecified. Yes, we should not overlook any potential mapping relationships. The preservation of length asymmetry during inference (Table 6) confirms that the framework successfully internalises such latent mapping rules; this was precisely the motivation for constructing LMT_{up}, LMT_{down}, and LMT_{eq}. After all, nothing demonstrates more intuitively than length, and LMT has indeed learned the mapping logic between primer sets and target sets. Consequently, the primer set operates as a steerable coordinate system in latent space: adjusting its properties allows users to navigate the generative results in a fine-grained, sample-specific manner. Our contribution centres on the learning framework itself; we have not pursued incremental mathematical refinements within each module, because nearly all the basic framework is explicitly designed to permit plug-and-play component substitution. We have systematically articulated the motivation, functionalities, components, execution protocol, empirical performance, and inherent advantages of the proposed learning framework. The primary objectives have been fully attained. On the other hand, we believe that how a method is applied matters more. The utility of a method is determined less by micro-optimisations than by the efficacy with which it is deployed. For instance, attention mechanisms predate Transformers, yet Transformers better harness this mechanism to construct their algorithmic models.

---

> ### Author Response · Authors · 2025-12-03
> **Response to 6HzQ (2/2)**
>
> W3. Thanks a lot. Your comment helpfully exposes a potential misreading. None of the three baseline models underwent retraining; they were invoked exactly as released by their respective authors, without modification. The sole retraining exercise concerned the LMT framework, whose reliance on a mapping table necessitates retraining whenever that table is altered. The original sentence "retrain ... ourselves" applies strictly to LMT.  We will amplify this clarification in the revised manuscript to preclude any ambiguity. Thus, it means that the training datasets for these three baselines are all richer than that of LMT. We do not hold any advantage at the dataset level. On the contrary, LMT is at a disadvantage in terms of training data.
>
> W4. Thank you for your comment. We have incorporated an expanded part in the revised manuscript (Sections 2.2 and 3.1). The converter is implemented as an encoder–decoder Transformer. Training employs label-smoothing KL divergence (LabelSmoothingKLDivLoss). Optimisation is performed with Adam encapsulated within the NoamOpt schedule: the learning rate is set to $(512^{-0.5} \cdot min(step^{-0.5}, step \cdot 4000^{-1.5}))$. The regularisation terms are dropout (p = 0.1) and label smoothing (0.1). Tokenisation is 20 standard amino acids, augmented with 3 special tokens (range: [0,22]).
>
> W5. We argue that numerous studies have successfully mined AMPs from diverse datasets by means of these three predictors. For instance, Xu et al. [6] mined AMPs from sludge, and Chen et al. [7] mined AMPs from the global marine microbial database. Indeed, all of the authors restricted themselves to employing these three predictors exclusively as predictive engines, without invoking any additional AI models. Thus, these models are widely acknowledged to constitute the present generation of premier predictors. I'm afraid that any predictor, no matter how impeccably engineered it is, remains intrinsically biased. Wet-lab validation thus remains the sole unassailable criterion of biological veracity. However, ICLR has recently accepted a series of studies confined entirely to purely in silico verifications [1-5], exemplified by the antibody design work of Wu et al. [3]. These standards inspired our own submission.
>
> [1] CBGBench: Fill in the Blank of Protein-Molecule Complex Binding Graph. ICLR, 2025.
> [2] Steering Protein Family Design through Profile Bayesian Flow. ICLR, 2025.
> [3] A Simple yet Effective $\Delta\Delta G$ Predictor is An Unsupervised Antibody Optimizer and Explainer. ICLR, 2025.
> [4] ReNovo: Retrieval-Based \emph{De Novo} Mass Spectrometry Peptide Sequencing. ICLR, 2025.
> [5] Fine-Tuning Discrete Diffusion Models via Reward Optimization with Applications to DNA and Protein Design. ICLR, 2025.
> [6] Waste to resource: Mining antimicrobial peptides in sludge from metagenomes using machine learning. Environment International, 2024.
> [7] Global marine microbial diversity and its potential in bioprospecting. Nature, 2024.
>
> W6. We release the source code in Supplementary Materials.
>
> Thank you again for your time and comments.

---

### Official Review · Reviewer_pBSX · 2025-11-01

**Soundness:** 2
**Presentation:** 1
**Contribution:** 2
**Rating:** 2
**Confidence:** 3

**Summary:**

This paper introduces Lightweight Midas Touch (LMT), a paired-learning framework that addresses challenges in antimicrobial peptide (AMP) generation by decoupling the traditionally tight coupling between training data and generative models. LMT augments this architecture with four key constituents: a primer set, a target dataset, an explicit mapping table, and a lightweight transformer-based converter. The framework operates on the principle that by liberating the primer set and mapping table from data-model constraints, practitioners gain modular control over generation through two mechanisms: (1) during training, different mapping tables produce distinct converters from the same target dataset, enabling exploration of diverse generative pathways, and (2) during inference, customized primers steer candidate generation without model retraining. Comprehensive evaluations on the APD3 database demonstrate that LMT achieves markedly superior diversity while maintaining competitive accuracy, effectively dissolving the inherent diversity-accuracy trade-off that plagues existing approaches.

**Strengths:**

1. This paper proposes a new pipeline for peptide generation, which consists of the primer set, the target set, the converter, and the mapping table. This converter can provide a more interpretable mapping between the primer and target peptide.
2. It assesses the model performance from multiple aspects to evaluate its effectiveness in generating diverse and accurate AMP. It also conducts a detailed study on the impact of the mapping table and test primers.

**Weaknesses:**

1. Some claims lack enough evidence. For example, Lines 110-118 mention the functional fixation challenge, which says that an unconstrained generated model cannot support conditional generation. However, VAE-based methods, such as HydrAMP, support both unconstrained generation and conditional generation. For unconstrained generation, they sample random noise as the input. For conditional generation for attributes, they can shift the latent space to meet the attribute. For conditional generation for a given input peptide, they can optimize the input by encoding it to an embedding with the VAE encoder and then shifting the latent representation.
2. While the introduction describes three challenges, it is not clear to me how these challenges are solved by the proposed method. LMT is still limited by the small AMP databases because it uses the AMPs as the target set. Although the converter can learn different mappings from the primer set to the target set, the target distribution is the same.
3. There is no guarantee that there is a learnable mapping for the converter, since the primer is randomly initialized and the mapping is stochastic. The mapping to learn may not reflect any meaningful underlying biological rules to build peptides. If that is the case, even if the mapping is learned, it cannot create meaningful peptides on a new prime set.
4. Figures and Tables
	1. Notation in Table 1 should be consistent with the main context. For example, Lines 357-358 say that the two models are Lt and LMT_ran, while in Table 1 they are LMT_tmp and LMP_ran.
5. Experimental results do not support the effectiveness of the proposed method in generating diverse and accurate peptides at the same time.
	1. In Table 1, the two primer sets have different performance in terms of diversity and accuracy. While one has a high diversity, its accuracy is low. The other has low diversity and high accuracy. There is no single primer that can outperform the other baselines in both diversity and accuracy. The sensitivity also shows that it is nontrivial to choose the right primer set, limiting the effectiveness and generalization of the proposed method
	2. In Table 2, the performance of different mapping tables further shows the sensitivity of the mapping, and more importantly, all these results have a low accuracy compared with baselines in Table 1. Similar problems happen in Table 3 as well, but for diversity.
6. A comprehensive discussion of related work is missing. Some related work is briefly mentioned in the introduction, but lacks a more detailed discussion. The related variants in the appendix are too general (diffusion model and language model).
	1. PepVAE: Variational Autoencoder Framework for Antimicrobial Peptide Generation and Activity Prediction
	2. PepCVAE: Semi-Supervised Targeted Design of Antimicrobial Peptide Sequences

**Questions:**

Based on the experiments and analysis, is there an empirical way to choose the test primers and mapping that can get a good balance between the diversity and accuracy?

---

> ### Author Response · Authors · 2025-12-03
> **Response to pBSX**
>
> Thank you for the time and effort spent on reviewing our work. We appreciate your comments and address your concerns and questions below.
>
> W1. We sincerely appreciate your perceptive comment, which has made us acutely aware that the generalized concepts "unconstrained" and "conditional" can obscure, rather than clarify. We will remove this description in the revised version to avoid such ambiguity. Thank you again.
>
> We now respond to the question about HydrAMP. The answer is straightforward. Once HydrAMP's framework is fixed, its functionality is fixed. However, its original algorithmic model simultaneously supports both unconstrained generation and template-similar generation (conditional generation).
> HydrAMP is built upon a cVAE. The model learns the distribution of the training data and can map sequences into a latent space. In this case, two operational modes are offered. 1) Unconstrained generation: latent variables are sampled directly from the prior distribution and decoded into novel sequences. 2) Template-similar generation (HydrAMP’s definition of “conditional”): a template sequence is encoded into a latent anchor point; a temperature-controlled neighbourhood around this anchor is sampled to yield sequences that are similar in distributional terms.
> If, however, "conditional generation" is construed more strictly, e.g., enforcing an exact match to the template's length or preserving its secondary-structure profile, HydrAMP is no longer adequate. Its latent-neighbourhood sampling mechanism lacks explicit modules for length or structural constraints; satisfying such requirements would necessitate architectural modifications such as a dedicated extra condition encoder or hard constraint layers. Thus, two levels exist: 1. The model's inherent capability (e.g., generating template-similar sequences); 2. Capabilities beyond the model's scope (e.g., length constraints, structural constraints). For HydrAMP, both unconditional generation and template-similar generation (conditional generation) remain confined to Level 1. Recall our viewpoint: The functionality of a model is fixed at the time of its construction. In summary, HydrAMP still follows this rule.
>
> W2. Thank you for your comment; the revised version includes an expanded description of this point (Section 2.4). Moreover, we must correct that the distribution recovered by any learner is never an exact replica of the true, unknown distribution; it is, unavoidably, a composite of the latter and the inductive bias introduced by the model. Data alone do not determine the generated results. Model and data jointly shape the estimated distribution. This observation is elementary, yet it cautions against idealized analyses that would render architectural innovation superfluous and meaningless. Returning to LMT, experimental results also demonstrate that LMT can indeed learn mapping relationships. The emergent distribution is co-determined by four factors: the primer set, mapping table, converter, and target set. Consequently, LMT facilitates the continual instantiation of different models, broadens the AMP repertoire, and thereby accelerates the discovery of novel candidates. Simultaneously, it allows existing AMP data to be exploited from multiple perspectives, enhancing data utility. It contributes at the data level.
>
> W3. We appreciate your insightful comment. In fact, the mapping is learned, as evidenced by Table 7 in Appendix D. We designed the explicit mapping tables and tested the generated sequences learned from them, denoted LMT_{eq}, LMT_{up}, and LMT_{down}, whose entries were randomly initialized, yet maintained a deterministic length correspondence between primer and target datasets. For instance, in the mapping table of LMT_{up}, the lengths of sequences in the target set exceed those of their corresponding primers.
> This readily quantifiable, length-explicit relation provides a direct probe for whether the mapping has been internalized: if the model failed to learn the mapping table, no systematic length association between primers and generated sequences would emerge at inference. The results in Table 7 confirm that models trained with the length-dependent mapping faithfully reproduce the prescribed relation during generation; sequences produced by LMT_up, for instance, are significantly longer than their corresponding primer sequences. These data demonstrate that LMT successfully acquires the mapping rule even when both the table and the primers are randomly constructed.

---

> > ### Author Response · Authors · 2025-12-03
> > **Response to pBSX (2/2)**
> >
> > W4. This appears to be a misunderstanding. First, L_t is the test primer dataset, while LMT_ran and LMT_tmp are generated sequences derived from the test datasets. These are fundamentally different concepts. We are testing the generated sequences, so only LMT_ran and LMT_tmp appear in Table1, and L_t cannot possibly be present in it. Furthermore, "LMP_ran" does not appear anywhere in the entire paper. I believe this is simply a typographical error by the reviewer. Even so, thank you for your comment.
> >
> > W5. Again, it appears to be a misunderstanding. In Table 1, LMT_{tmp} indeed outperforms all other baselines. First, LMT_{tmp} has the best accuracy. Second, the diversity of LMT_{tmp} exceeds all baselines. Note that APD3 is not a baseline but a training dataset.
> >
> > W6. We appreciate your insightful comment; accordingly, we have expanded Appendix A to discuss related work.
> >
> > Q1. LMT_{tmp} is one of the good examples to balance between the diversity and accuracy, as shown in Table 1.
> > In fact, LMT was originally designed to support flexible AMP design. This stems from our observation that existing protein generation models have fixed functionalities and cannot flexibly accommodate new requirements (a phenomenon likely common across AI for science applications). So why not design a new Framework that flexibly supports switching between new functionalities? Yes, the key point of LMT is that it is a flexible, switchable architecture, not confined to a single model. This makes the flexible design of AMPs possible. We only need to adjust mapping tables as needed. For instance, if the goal is purely unsupervised aAMP discovery, a simple random mapping table suffices. Experimental results show that LMT_{tmp} effectively balances precision and diversity. When working with promising sequence templates, mask-based mapping table constructions like LMT_{m02} can be explored. This approach tends to regenerate templates with antibacterial potential, and so on. The appeal of LMT lies in its extensive design flexibility, making it well-suited to adapt to diverse application needs. Consequently, this variability makes it challenging to guarantee optimal precision and diversity under all circumstances while simultaneously accommodating new requirements. We contend it is more of a universal principle rather than a defect peculiar to LMT.
> >
> > Thank you again for your time and comments.

---

### Meta-Review · Area_Chair_U78R · 2026-01-05

**Summary:**

The consensus for rejection is driven by critical concerns regarding the methodological soundness, validation validity, and experimental rigor of the proposed framework. Reviewers questioned the efficacy and novelty of the random mapping and paired-learning, noting that the approach essentially rebrands existing conditional generation paradigms without genuinely resolving the trade-off between diversity and accuracy. The evaluation wa insufficient due to its exclusive reliance on in silico deep learning predictors—which lack wet-lab verification and risk circular validation bia. While the experimental design faced scrutiny for unfair baseline comparisons, significant redundancy in generated sequences due to data augmentation, and the omission of stronger, established baselines like ProtGPT2.

**Reviewer Concerns:**

The rebuttal successfully addressed technical queries regarding reproducibility and baseline fairness by clarifying that the authors utilized baselines at a data disadvantage, adding a requested ProtGPT2 comparison, and demonstrating via empirical evidence that the "andom mapping effectively learns constraints like sequence length without relying on the redundancy-inducing data augmentation used in their supplementary analyses.

However, critical concerns regarding the validity of the evaluation and the fundamental novelty of the approach remain. Citing recent ICLR precedents failed to satisfy the reviewers' demand for biological verification or external benchmarking. The rebuttal did not effectively counter the methodological critique that the paired-learning framework functions essentially as a standard sequence-to-sequence model memorizing random codes, nor did it definitively prove that the intrinsic trade-off between diversity and accuracy has been algorithmically resolved rather than merely shifted through model selection.

**Reviewer Scores:**

The final scores remained at 2, 2, and 6 because the rebuttal failed to satisfy the fundamental methodological objections of the critical ones. Reviewers pBSX and 6HzQ maintained their scores of 2 because their core issues were defended by the authors rather than rectified, leaving the reviewers' skepticism regarding soundness and contribution intact. Reviewer WkGC maintained a score of 6. Since the author provided the requested ProtGPT2 baseline and missing implementation details, reviewers might not lower their score.

---

### Decision · Program_Chairs · 2026-01-26

Reject